# Predictive Factors for Resistant Disease with Medical/Radiologic/Liver-Directed Anti-Tumor Treatments in Patients with Advanced Pancreatic Neuroendocrine Neoplasms: Recent Advances and Controversies

**DOI:** 10.3390/cancers14051250

**Published:** 2022-02-28

**Authors:** Lingaku Lee, Irene Ramos-Alvarez, Robert T. Jensen

**Affiliations:** 1Digestive Diseases Branch, NIDDK, NIH, Bethesda, MD 20892-1804, USA; frkn505@gmail.com (L.L.); irene.ramosalvarez@nih.gov (I.R.-A.); 2National Kyushu Cancer Center, Department of Hepato-Biliary-Pancreatology, Fukuoka 811-1395, Japan

**Keywords:** pancreatic neuroendocrine neoplasms, prognostic factor, everolimus, sunitinib, PRRT, somatostatin analogue, chemotherapy

## Abstract

**Simple Summary:**

Tumor resistance, both primary and acquired, is leading to increased complexity in the nonsurgical treatment of patients with advanced panNENs, which would be greatly helped by reliable prognostic/predictive factors. The importance in identifying resistance is being contributed to by the increased array of possible treatments available for treating resistant advanced disease; the variable clinical course as well as response to any given treatment approach of patients within one staging or grading system, the advances in imaging which are providing increasing promising results/parameters that correlate with grading/outcome/resistance, the increased understanding of the molecular pathogenesis providing promising prognostic markers, all of which can contribute to selecting the best treatment to overcome resistance disease. Several factors have been identified that have prognostic/predictive value for identifying development resistant disease and affecting overall survival (OS)/PFS with various nonsurgical treatments of patients with advanced panNENs. Prognostic factors identified for patients with advanced panNENs for both OS/PFSs include various clinically-related factors (clinical, laboratory/biological markers, imaging, treatment-related factors), pathological factors (histological, classification, grading) and molecular factors. Particularly important prognostic factors for the different treatment modalities studies are the recent grading systems. Most prognostic factors for each treatment modality for OS/PFS are not specific for a given treatment option. These advances have generated several controversies and new unanswered questions, particularly those related to their possible role in predicting the possible sequence of different anti-tumor treatments in patients with different presentations. Each of these areas is reviewed in this paper.

**Abstract:**

*Purpose*: Recent advances in the diagnosis, management and nonsurgical treatment of patients with advanced pancreatic neuroendocrine neoplasms (panNENs) have led to an emerging need for sensitive and useful prognostic factors for predicting responses/survival. *Areas covered*: The predictive value of a number of reported prognostic factors including clinically-related factors (clinical/laboratory/imaging/treatment-related factors), pathological factors (histological/classification/grading), and molecular factors, on therapeutic outcomes of anti-tumor medical therapies with molecular targeting agents (everolimus/sunitinib/somatostatin analogues), chemotherapy, radiological therapy with peptide receptor radionuclide therapy, or liver-directed therapies (embolization/chemoembolization/radio-embolization (SIRTs)) are reviewed. Recent findings in each of these areas, as well as remaining controversies and uncertainties, are discussed in detail, particularly from the viewpoint of treatment sequencing. *Conclusions*: The recent increase in the number of available therapeutic agents for the nonsurgical treatment of patients with advanced panNENs have raised the importance of prognostic factors predictive for therapeutic outcomes of each treatment option. The establishment of sensitive and useful prognostic markers will have a significant impact on optimal treatment selection, as well as in tailoring the therapeutic sequence, and for maximizing the survival benefit of each individual patient. In the paper, the progress in this area, as well as the controversies/uncertainties, are reviewed.

## 1. Introduction

Unfortunately, a significant percentage of patients with both functional panNEN syndromes [insulinoma, Zollinger-Ellison syndrome (ZES), other malignant functionalpanNENs) (F-panNEN) and nonfunctional panNENs (NF-panNENs) cannot be cured by surgical resection at presentation, or develop advanced disease on follow-up, which is not surgically curable, and which markedly reduces their survival and requires anti-tumor therapies [1,2,3,4,5,6,7,8,9,10]. The prognosis in patients with advanced pancreatic neuroendocrine neoplasms (panNENs) has been generally poor and, until recently, there has been little improvement, mainly due to the lack of new therapeutic approaches which have led to effective anti-tumor treatment activity in these patients [7,9,11,12,13,14,15]. In the past decade, however, there has been significant advances, not only in the treatment approaches for panNEN patients with advanced disease, the recognition into the pathogenesis of the acquired and primary resistance to anti-tumor treatment which is an increasing problem, but also in the diagnosis, management and overall treatment of panNENs, as well as other NENs, which is beginning to influence survival rates in these patients [7,11,14,15,16,17,18,19,20,21,22]. We have recently analyzed both these advances and controversies in identifying prognostic factors for the overall management of panNEN patients, as well as for surgical outcomes [7]. In the present paper, we have extended this analysis to review the current identification of predictive/prognostic factors for the other nonsurgical therapeutic modalities used in the treatment of patients with panNENs with advanced disease, as well as the identification of resistance to treatment limiting their efficacy.

In terms of recent changes in the nonsurgical treatment of patients with advanced panNENs, several recent pivotal Phase 3 trials have markedly changed management. Two Phase 3 studies showed significantly prolonged progression-free survival (PFS) of panNEN patients with advanced disease treated with new medical treatments: one with everolimus (mammalian target of rapamycin [mTOR] inhibitor) [23] and the other with sunitinib (a multikinase tyrosine inhibitor) [24], compared with treatment with placebo, and these agents are now globally used as key drugs in the treatment of advanced panNENs [5,9,16,25,26,27,28,29,30,31,32]. Recently, two additional Phase 3 studies have shown the anti-tumor effectiveness of another multityrosine kinase receptor inhibitor, surufatinib [33] in both advanced panNENs [34] and extrapancreatic NENs (carcinoids) [35]. PanNENs ectopically overexpress somatostatin receptors type 1–5 (SSTR 1–5) and numerous studies showed their activation has anti-tumor and anti-secretory effects in NENs [22,36,37,38,39,40,41,42,43,44]. Like recent studies demonstrating that the somatostatin analogue (SSA) octreotide had an anti-tumor effect extending PFS in ileal carcinoids and other tumors [22,37,38,40,45], the long acting SSA lanreotide demonstrated similar anti-tumor activity in a recent Phase 3 trial involving patients with advanced gastroenteropancreatic NENs (GEP-NENs), in addition to its having an anti-secretory effect [22,37,46]. PanNENs as well as NENs in other locations and many more common neoplasms frequently ectopically overexpress various G-protein coupled receptors, and these are increasingly being used to not only localized these tumors using various receptor ligands, as well as for targeted delivery of cytotoxic compounds [47,48,49,50,51,52,53]. The overexpression of somatostatin receptors by panNENs and other NENs is also being used to treat advanced disease using radiolabeled somatostatin analogues [25,40,42,54,55,56,57]. The efficacy of peptide receptor radionuclide therapy (PRRT) using ^177^Lutetium (Lu) -DOTATATE was also reported from another Phase 3 trial involving patients with advanced midgut NENs [58] and from studies of patients with advanced panNENs primarily from Rotterdam [59]. The results from these trials led to the approval for PRRT use both in the US and in other countries. These results are increasingly supported by additional recent non-prospective studies [25,56,57,60,61,62,63]. Several studies have reported the anti-tumor efficacy of various chemotherapies [13,14,18,31,64,65,66,67,68], as well as liver-directed therapies including transarterial chemoembolization/embolization (TACE/TAE) or radio-embolization [14,18,31,69,70,71,72,73,74].

Despite these recent advances, resistance to anti-tumor therapies, either primary resistance or acquired resistance, in patients with advanced PanNENs, is an increasing problem [6,20,21,29,75,76,77,78,79]. This is a similar finding to its occurrence with similar anti-tumor treatments in patients with other non-NEN advanced malignancies [80], as well as patients with advanced NENs in other non-pancreatic locations (i.e., carcinoids) [6,29,76,77,78,79]. While the acquired resistance to therapy with targeted molecular therapies (i.e., mTOR inhibitors, Tyrosine kinase inhibitors) in patients with advanced panNENs/other NENs has received particular attention [6,21,29,75,77,78,81]; both primary and acquired resistance occur with the other forms of nonsurgical anti-tumor therapies in these patients to varying degrees (Figure 1). As is apparent from this figure (Figure 1), a significant degree of resistance is seen with all therapies, including somatostatin analogues, PRRT, chemotherapy and targeted therapy. Furthermore, in most cases a significant degree of resistance is seen within 1–2 years of treatment (Figure 1). The early identification of patients that will demonstrate subsequent resistance to a given anti-tumor treatment would be of great clinical value in not only allowing earlier use of another therapy that might be successful, but in also in helping to select which alternate therapy might be the best.

The increasing number of these available treatment options for patients with advanced panNENs has also led to controversies and uncertainties about the exact role and current position of each of these therapeutic agents in the multimodal therapeutic approaches for these patients [7,9,56,61,73]. Despite the significant risk reduction of disease progression in patients with advanced panNENs demonstrated with the use of everolimus, sunitinib and lanreotide, their direct effect on overall survival/prognosis remains unclear [7,23,24,29,40,46], primarily due to the study design of these Phase 3 trials that used PFS as the primary endpoint and allowed placebo-treated patients to receive open-label use of active drugs. Recent meta-analyses/reviews including previous randomized control trials [61,102,103] of treatments of patients with advanced panNENs/carcinoids, as well as a nationwide population study in the Netherlands [104], suggest a higher survival benefit with PRRT treatment than the other systemic therapies in these patients. A second recent meta-analysis [105] compared active surveillance, which is recommended for patients initially diagnosed with metastatic NENs in several guidelines, to active antitumor treatment (somatostatin analogues, sunitinib, everolimus, PRRT), and found that active treatment extended to both PFS and OS. Another recent meta-analysis comparing the cytoreductive effect between different therapeutic agents reports chemotherapy alone (capecitabine/temozolomide) or in combination showed the strongest effect of cytoreduction in patients with advanced panNENs, followed by PRRT and sunitinib [106]. While everolimus, SSAs and sunitinib prolong PFS in patients with panNENs/NENs, in most cases they have a tumor-stabilizing effect rather than a tumor reduction effect [16,23,24,29,37,40,45,46,107]. The above results suggest that there may be major survival differences in the different medical/radiological therapeutic treatment modalities, as well as different safety/side-effect profiles with the different treatments. There has been no prospective, randomized controlled study investigating the efficacy of PRRT in panNENs only, although its high anti-tumor activity was reported from a retrospective database involving many panNEN patients [59], as well as in other non-prospective studies [37,38,55]. Most studies reporting the efficacy of chemotherapies and liver-directed therapies were performed more than a decade ago in the form of single-arm, non-randomized, retrospective studies in a small cohort of patients, and thus their exact roles in the treatment sequence including various recently-developed therapeutic options remain uncertain [108]. In panNEN patients with G3 NEC (poorly-differentiated) tumors, platinum-based chemotherapy, primarily the combination of cisplatin and etoposide, is recommended as first-line therapy, whereas possible options for second-line treatment include temozolomide-, irinotecan- or oxaliplatin-based regimens [66,67]. However, the true efficacy of chemotherapy in patients with G3 NEC tumors is presently unclear [109]. In addition, the exact role of PRRT in the treatment of both G3 NETs and G3NECs, at present, is not clear [110,111,112].

Recent advances in imaging and diagnostic modalities have significantly changed the management and treatment of panNENs but have also introduced some controversies and uncertainties [7,47,48,113]. The availability of radionuclide imaging modalities and sensitive biomarkers result in higher detection rates of metastatic lesions, as well as more accurate and earlier detection of disease progression than was available with traditional imaging/tumor localization methods including cross-sectional imaging (ultrasound, CT, MRI), angiography or measuring of hormonal gradients [47,48,114,115,116,117,118,119,120,121,122]. Changes in imaging modalities are not only important for improved localization of tumor site/extent, but also are becoming particularly important for their ability to change clinical treatment/management, not only by better assessing disease extent, but also by the development of radiological prognostic factors [47,48,123,124,125], including from texture analyses of the imaging results, various parameters from image tumor contrast patterns, computation of imaging modalities SUV/Max and other isotope parameters [47,48,126,127].

Numerous blood/tumor markers have been proposed to be of use for determining potential responsiveness to therapies as well as resistance, but their exact use is associated with controversies/uncertainties. Recently, gene transcript analysis using the NETest, a blood-based multianalyte NET gene signature, or assessment of other circulating NET signatures, is receiving increasing attention as a possible useful biomarker in diagnosis and/or for monitoring results of various anti-tumor therapies in patients with panNENs, as well as other NENs [128,129,130,131,132,133,134,135,136,137,138], in addition to the widely used other blood general biomarker, chromogranin A (CgA) [139,140,141,142,143,144,145,146,147,148,149,150,151,152,153] and various specific F-panNEN markers [1,90,91,92,154,155,156,157,158,159,160,161,162,163,164]. At present, whether the NETest or assessment of other circulating NEN transcripts can be used as a screening test or as a marker to monitor treatment response remains unclear, with a recent study reporting its limited diagnostic value due to a low specificity [130]. Therefore, there remain several controversies in this area about which diagnostic/prognostic test to use, when they should be used, as well as how to analyze as well as how to interpret the data [7]. These controversies/uncertainties will be dealt with in more detail in the latter sections of the paper that deal with the discussion of prognostic factors for assessing responses to different specific anti-tumor therapies.

The above controversies and uncertainties underlying the treatment/management of patients with advanced panNENs, especially related to the increasing role of resistant to treatment, have led to the emergence of a need for the development/establishment of sensitive and useful prognostic markers predictive for the clinical effects/outcomes of each specific therapeutic option for several reasons (Table 1). The availability of such prognostic markers will help clinicians select the optimal therapeutic agent at a particular time and to tailor treatment sequences for each individual patient, because the anti-tumor activity and toxicity can vary markedly between each therapeutic option, resulting in different clinical outcomes [18,25,26,61,73,102,103,104,106]. In addition, due to the lack of standardized treatment/follow-up protocols for some therapeutic options, factors predicting response to these treatments are becoming increasingly important in tailoring treatment schedules/sequence, considering combination and/or conversion therapy, as well as how to follow-up with these patients. Furthermore, predictive factors on primary/acquired resistance to each therapeutic option, as well as the occurrence of severe/unfavorable treatment-related adverse events (AEs), can also have a significant impact on treatment decision making [27,29,54,73].

To address these issues, several recent studies have reported the potential of various clinicopathological and genetic/molecular factors, as well as biomarkers, in predicting therapeutic response and prognosis to each of these available nonsurgical therapeutic options in patients with advanced panNENs. However, at present, evidence that supports the clinical routine use of these prognostic/predictive factors in the treatment and management of individual patients with advanced panNENs is still lacking. Furthermore, these proposals for their use have generated both controversies and uncertainties. Therefore, in this paper we review and summarize the recent insights of various prognostic markers predictive for efficacy of these available nonsurgical therapeutic agents, particularly in identifying those patients who demonstrate progressive disease resistant to a given therapy. We have done this by analyzing the available literature on predictive factors during various nonsurgical treatments for patients with advanced panNENs for both overall survival and disease-related survival. We have also included a discussion about emerging controversies and uncertainties in these areas.

## 2. Methods

This paper reviews and discusses in patients with advanced panNENs the predictive value of clinically-related factors (clinical, laboratory, imaging, treatment-related factors), pathological factors (histological factors/classification/grading), and molecular factors on clinical outcomes for each anti-tumor therapeutic modality including molecular targeted therapies (everolimus, sunitinib and SSAs), radiological SSTR-targeting therapy-PRRT, chemotherapy for well- and poorly-differentiated tumors, and liver-directed therapies (TACE/TAE and radio-embolization). We summarized the recent insights from the studies within the last five years, concentrating primarily on studies that are published from the past three years which are available in MEDLINE, abstracts from meetings, or meeting proceedings, and which provide prognostic/predictive information (progression-survival [PFS] and overall survival [OS]). This paper also reviews in detail the emerging controversies/uncertainties in these areas in a separate section. Factors affecting only the radiological response (i.e., objective response rate [ORR] and disease control rate [DCR]), and toxicities related to each treatment will only be discussed in brief in this review because of the limited space. This review primarily concentrated on reports consisting of patients with panNENs (pancreas origin), but also includes reports that have a large proportion of panNEN patients. However, in some sections which include a small number of studies of patients with panNENs only, we have included results from studies, even if the proportion of panNEN patients is small. 

## 3. Predictive Factors for Response to SSA in Advanced panNENs

### 3.1. General: Predictive Factors with Somatostatin

Currently, there are two long-acting SSAs which are widely used for panNENs/NENs: octreotide LAR and lanreotide autogel [9,40]. Numerous studies in animals, isolated cells and non-prospective/non- randomized studies in humans provide evidence that both octreotide and lanreotide have NEN anti-tumor activity [9,22,37,38,40,44,107,165]. Besides the significant anti-secretory effect of these SSAs, lanreotide is the only agent which is approved to use for its anti-growth effect in panNENs, however, both are globally used for their anti-tumor effects in gastrointestinal (GI)-NENs [37,165]. While the approved indications for these two long-acting SSAs differ, the National Comprehensive Cancer Network (NCCN) guideline considers both drugs to be appropriate interventions for symptom control and delay of GEP-NEN progression [166]. Despite its weak cytoreductive effect, the mild toxicity of SSAs as shown in a recent study with lanreotide having a low occurrence of severe AEs (26%) and preserved quality of life [46] have led to the recommendations for their initial use in patients with advanced panNENs proposed in numerous guidelines/expert reviews [5,37,38,40,166,167,168,169]. However, long-term use of SSAs can result in the development of occasional AEs that require treatment, specifically biliary stone diseases [170] and metabolic disorders [171].

Like PRRT, most recent studies investigating prognostic factors for SSA treatment included a proportion of other NENs, particularly GI-NENs, and therefore, in this section, we will discuss the recent findings reported form the studies including patients with all GEP-NENs. For previous findings regarding predictive factors on SSA efficacy from older studies, see [165].

### 3.2. Clinically-Related Predictive Factors for SSA Efficacy [Clinical, Laboratory, Treatment-Related Factors]

Like the findings observed with PRRT, poorer performance status [172], symptomatic tumors [173] and the prior lack of primary tumor resection [172] are associated with decreased PFS with SSA treatment (Table 2, Right Panel). Two different studies report significant decreases in PFS in patients with baseline (pre-SSA) documentation of disease progression [173,174], suggesting a significant but mild anti-tumor activity of SSAs in patients with panNENs, as well as with GI-NENs. In this regard, an exploratory analysis of the Phase 3 CLARINET study emphasized the importance of calculating tumor growth rate at baseline in predicting disease progression/mortality with lanreotide treatment [174].

Several studies report a worse clinical outcome in patients with elevated baseline levels of CgA [172,175], whereas the decrease in CgA levels after SSA treatment is predictive of an improved PFS [176,177]. A recent post hoc analysis of the Phase 3 CLARINET study demonstrated that 5-HIAA responders had significantly improved PFS with lanreotide treatment (*p* = 0.007), while no significant difference in PFS was observed between CgA responders and no-responders. In addition, the predictive utility of the NETest on disease progression is reported from a prospective study involving 28 patients with GEP-NENs (9 panNENs) [178]. An increase in the NETest value occurred significantly earlier than radiological documentation in patients with disease progression, as well as than CgA alterations [178]. A recent study involving 535 patients with GEP-NENs (177 panNENs) reports that higher neutrophil-to-lymphocyte ratio and alkaline phosphatase levels were significantly associated with decreased PFS [173].

Like that observed with everolimus treatment, longer times to disease progression and longer PFS with SSA treatment were both associated with longer OS [175]. This correlation was stronger in patients with elevated CgA levels, as well as with functional tumors, although these differences were not statistically significant [175]. In addition, the presence of prior systemic therapy is predictive of both deceased PFS [179,180] and OS [179] with SSA therapy. 

### 3.3. Pathological Predictive Factors for SSA Efficacy [Histological Factors/Classification/Grading, Molecular Factors]

An increased tumor grade [42,172,179,223], presence of a poorly-differentiated tumor [175] and an increasing Ki-67 index [173] are reported to correlate with worse outcomes with SSA treatment (Table 2, Right Panel). Some studies reported that a Ki-67 threshold of 5% was more predictive of PFS with SSA treatment than the cut-off value proposed by 2010 WHO classification to separate G1 and G2 (i.e., <3%) [172,224]. SSA therapy is less effective in patients with bone and peritoneal metastases [173], as well as with any liver metastases [42] or a high hepatic tumor load [173,174,177,179,225], suggesting its limited role in advanced panNEN patients with aggressive tumors. 

A recent study involving 52 patients with NENs (20 panNENs) reports a significant correlation between high disease control rate and positivity of SSTR2A expression (*p* = 0.045), whereas assessment of SSTR5 expression was not predictive for response to SSA treatment [226].

## 4. Predictive Factors for Response to Everolimus in Advanced panNENs

### 4.1. General: Predictive Factors with Everolimus

According to the pivotal everolimus Phase 3 trial (RADIANT-3) involving patients with advanced panNENs, everolimus prolonged PFS (11.0 months) and OS (44.0 months) over that observed in placebo-treated patients (PFS 4.6 months, OS 37.7 months) [23,90] (Table 3). A recent meta-analysis of everolimus use in advanced neuroendocrine tumors reported a significant beneficial effect by increasing PFS but showed no difference for overall survival [32]. However, the results from numerous studies investigating the efficacy of everolimus report that up to 57% of patients experienced disease progression, primarily due to resistance to everolimus [29] (Figure 1). The efficacy of everolimus was characterized by high disease stabilization (DCR, 73%), but not by its cytoreductive effect (ORR, 5%) [23]. Numerous studies have investigated the efficacy of various anti-tumor agents, specifically SSAs, in combination with everolimus, however, none of these showed a significant synergistic effect in the previous prospective clinical trials [29].

### 4.2. Clinically-Related Predictive Factors for Everolimus Efficacy [Clinical, Laboratory, Treatment-Related Factors]

Recently reported independent predictors for decreased PFS/OS with everolimus treatment include poorer performance status [228,229] and the prior lack of primary tumor resection [231,232] (Table 3, Left Panel). Other studies report that patients who achieved tumor growth control had longer OS [94], and longer times to disease progression and longer PFS were both associated with improved OS [175]. It is reported that the use of higher cumulative everolimus doses (>3000 mg) are significantly associated with longer PFS and OS, whereas low everolimus dosing (<9 mg/day) correlates with shorter PFS [243]. On the other hand, dose reduction/interruption are frequently required to control the severity of AEs of everolimus, raising the importance of therapeutic drug monitoring [29]. A recent multicenter study in Italy involving 445 panNEN patients receiving everolimus and/or SSA reported significantly longer PFS in diabetic patients compared to that observed in non-diabetic patients (HR, 0.63; *p* = 0.0002) [231]. The concomitant use of metformin in this study was significantly associated with improved PFS [231], while there was no longer a significant difference when the results only in patients treated with everolimus alone were analyzed due to the small number of patients (*n* = 37). These results suggest that there is a synergistic anti-tumor effect of metformin on everolimus and/or SSA, at least partially by reinforcing mTOR inhibition and suppressing the IGF-1 oncogenic axis [231]. This important possibility will need to be validated by future prospective, randomized studies.

In terms of biomarkers predictive for everolimus’s effectiveness, an exploratory analysis of the Phase 3 RADIANT-3 trial showed that patients with elevated basal levels of chromogranin A (CgA), neuron specific enolase (NSE), placental growth factor, and soluble vascular endothelial growth factor-1 (VEGFR1) had significantly shorter OS [90]. The predictive value of these biomarkers in response to everolimus treatment is also reported in a sub-analysis of the Phase 2 RADIANT-1 study, which showed that baseline levels of CgA and NSE, as well as their early biochemical response, are independent predictors for PFS and OS [228,233]. In addition, a pooled analysis of RADIANT-3/4 reported that markers of systemic inflammation, neutrophil-to-lymphocyte ratio (NLR) and lymphocyte-To-monocyte ratio (LMR), are predictive of PFS [234].

The development of specific everolimus-related AEs, including hypercholesterolemia [232] and stomatitis [244], is reported to be associated with a favorable clinical outcome with everolimus treatment. These can help predict therapeutic response even after the initiation of everolimus, while the predictive value of the development of hypercholesterolemia was not observed in a pooled analysis of RADIANT-3/4 [245]. A sub-analysis of RADIAT-3 trial showed that previous chemotherapy did not affect the efficacy and toxicity of everolimus treatment [246], and another study also reported similar efficacy and toxicity of sequential treatments with everolimus and sunitinib [101]. In contrast, a study involving 169 patients with NENs (85 panNENs) reported previous chemotherapy and PRRT as significant risk factors for the development of severe AEs, including increased hematological toxicity, renal failure, peripheral edema, pneumonitis and mucositis [94].

### 4.3. Pathological Predictive Factors for Everolimus Efficacy [Histological Factors/Classification/Grading, Molecular Factors]

Specific histological factors are predictive for response to everolimus treatment, particularly in panNEN patients. Similar to the results with overall prognosis and post-survival outcomes as described previously [7], patients with higher tumor grade [95,230,231,232] and distant metastases [231,232,238] are reported to have worse outcomes with everolimus treatment (Table 3, Left Panel).

In terms of predictive molecular factors for everolimus treatment efficacy, a recent study involving 58 patients with well-differentiated panNENs demonstrated that higher intra-tumoral expression of acetyl-CoA carboxylase 1 (ACC1), a key enzyme in fatty acid biosynthesis, is significantly associated with decreased PFS, suggesting the importance of tumor lipid metabolism in everolimus treatment [237]. Another recent study reported a negative correlation between the expression level of phosphorylated p70S6K, a downstream effector of mTOR pathway, and PFS/OS [232]. However, previous studies reported conflicting results regarding the predictive value of assessing the genetic/protein expression of the PI3K/mTOR pathway on clinical outcomes with everolimus treatment [29,247], thus this association still needs further validation in future studies.

According to a genomic study of a subset of RADIANT everolimus trials [248], patients with high chromosomal instability (CIN) showed a trend toward longer OS (HR, 0.55; *p* = 0.077), even after adjusting for baseline CgA/NSE levels (HR, 0.53; *p* = 0.068). Extended OS in the CIN high subgroup is also reported with the low baseline levels of both CgA (HR, 0.43; *p* = 0.058) and NSE (HR, 0.6; *p* = 0.21), whereas CIN status did not affect PFS [248]. In addition, either the presence of MEN-1 (Multiple Endocrine Neoplasia-type 1) mutations, or mutations of DAXX (death-domain-associated protein) or ATRX (alpha thalassemia/mental retardation syndrome X-linked), did not improve OS with everolimus treatment (*p* = 0.34) [248]. The treatment-specific effect of mutations in the PI3K pathway genes could not be evaluated in this study, as most of these patients received placebo [248]. PanNENs are known to occur with increased frequency in a few inherited syndromes [MEN1 > VHL > neurofibromatosis > tuberous sclerosis] [249,250,251,252], are frequently multiple, and can be malignant and a leading cause of death (MEN1) [157,253,254,255]. One small retrospective study (33 patients, 8 with MEN1/VHL) of patients with metastatic panNENs reported the time to tumor progression and PFS were numerically higher in patients with germline mutations than those with sporadic disease treated with everolimus [256]. For further information from older studies, as well as predictive markers of everolimus in other malignancies, see [29,247,257].

## 5. Predictive Factors for Response to Sunitinib in Advanced panNENs

### 5.1. General: Predictive Factors with Sunitinib

In a previous pivotal Phase 3 trial in panNEN patients, sunitinib demonstrated significant risk reduction for disease progression compared to placebo, with a PFS of 11.4 months and an OS of 38.6 months, respectively, compared to placebo with PFS of 5.5 months and OS of 29.1 months, respectively [24,258] (Table 3, Right Panel). Like everolimus, treatment with sunitinib showed a high degree of disease stabilization (disease control rate [DCR], 72%) with tumor shrinkage uncommonly occurring (ORR, 9.3%) [24], whereas even higher DCR (24.5%) was reported in a recent Phase 4 study [258]. At present, the anti-tumor efficacy of sunitinib in NENs is only proven in patients with panNENs. 

The most frequent AEs related to sunitinib treatment include abdominal pain, stomatitis, hypertension, hand-foot syndrome, bleeding and hypothyroidism [24,27,30,258]. Dose/schedule adjustments are frequently required to manage sunitinib-related AEs; however, its effect on the anti-tumor activity of sunitinib remains controversial [28,259,260]. 

### 5.2. Clinically-Related Predictive Factors for Sunitinib Efficacy [Clinical, Laboratory, Treatment-Related Factors]

In a post hoc analysis of previous Phase 3 and Phase 2 studies involving 152 patients treated with sunitinib, a univariate analysis revealed a 10–30% reduction in the size of marker lesions on imaging as a significant predictor for improved PFS, whereas only the threshold of 10% remained statistically significant after multi-variate analysis was applied [239] Table 3, Right Panel). Similarly, in a sunitinib treatment study involving 18 patients with GEP-NEN (14 panNENs), there was a significantly longer time to progression in patients who experienced a partial imaging response [240]. In this study, the authors emphasized the importance of assessing the change in tumor vascularization/density (i.e., Choi criteria) in predicting response to sunitinib treatment, rather than change in tumor diameter (i.e., RECIST criteria) [240]. A similar conclusion was reached in a recent study of sunitinib treatment of 107 patients with advanced panNENs with Choi criteria better correlating with OS and PFS than RECIST criteria [28]. In addition, a recent Phase 2 sunitinib study in France involving 31 patients with grade 3 GEP-NENs (13 panNENs), reported that patients who achieved disease control had significantly longer OS than non-responders (*p* = 0.001), while no difference was observed with PFS (*p* = 0.89) [242]. These results support the survival benefit of sunitinib in patients with panNENs, regardless of tumor grades.

In terms of circulating biomarkers, an exploratory analysis of the Phase 2 sunitinib study reported that high baseline levels of soluble VEGFR-2 predicted longer OS (*p* = 0.01) in 66 patients with advanced panNENs [235]. In a subset of 28 patients with panNENs (*n* = 14) or carcinoid tumors (*n* = 14), high levels of stromal cell-derived factor (SDF)-1α correlated with an increased risk of progression (*p* = 0.005) or mortality (*p* = 0.02) [235].

In addition, genomic analyses from two different Phase 4 trials examined the predictive value of genetic profiles using blood samples, specifically genes related to the VEGF pathway, on clinical outcomes with sunitinib treatment [236,261]. According to a prospective Phase 4 study in Spain including 43 patients with G1/2 panNENs, two single-nucleotide polymorphisms (SNPs) in the VEGFR3 gene detected from blood samples were significantly associated with decreased OS (rs307826; *p* = 0.01, rs307821; *p* = 0.005) [236]. Among 6 circulating biomarkers examined in the same study, interleukin-6 was associated with poorer OS (*p* = 0.013), whereas osteopontin was associated with decreased PFS (*p* = 0.023) [236]. Similarly, a genomic sub-analysis from another Phase 4 sunitinib study including 56 patients with well-differentiated panNENs demonstrated a trend toward shorter PFS in patients with SNPs on VEGFR1 rs9554320 (*p* = 0.117) and VEGFR1 rs9582036 (*p* = 0.102) [261]. Correlations between objective response rate (ORR) and SNPs of VEGFA rs2010963 and rs833068, VEGFR1 rs9582036 and VEGFR2 rs7692791 are also reported, although they were not statistically significant [261]. In contrast, IL-1β SNPs was significantly associated with higher ORR (*p* = 0.001) [261]. Therefore, the assessment of genetic profiles, particularly genes related to VEGF pathway, as well as certain circulating biomarkers, can have important clinical implications for optimal treatment decision making, although larger prospective studies are required to validate these findings.

In terms of treatment-related factors (Table 3, right panel), results from a recent Phase 4 study in panNEN patients treated with sunitinib show comparable anti-tumor activity and toxicity profiles between treatment-naïve patients and previously treated patients [28].

The results from an extension study of a Phase 3 sunitinib trial reported the significant correlation between several risk factors with the occurrence of several AEs, including a higher risk of hypertension with poorer baseline performance status (≥1) and history of prior hypertension, hand-foot syndrome with race (non-white), and history of diabetes, bleeding with long-term use of sunitinib, and cardiac AEs with higher age (≥65), respectively [27]. A recent Phase 4 study also reports that a higher rate of dose reduction was required with the presence of SNP of VEGFR3 rs307826, whereas the SNP VEGFR3 rs307821 was associated with the higher occurrence of hypothyroidism [236]. These can also affect the indication as well as the management of AEs with sunitinib treatment.

### 5.3. Pathological Predictive Factors for Sunitinib Efficacy [Histological Factors/Classification/Grading, Molecular Factors])

There is minimal evidence regarding histological factors for predicting outcomes with sunitinib in panNEN patients, except for several studies reporting significantly worse PFS/OS with higher Ki-67 index [227,236] and higher mitotic index [227] (Table 3, Right Panel). A significantly shorter PFS and OS with sunitinib treatment are also reported in patients with G3 NEN tumors than with G1/2 NET and G3 NET tumors, whereas there was no difference in PFS/OS between patients with G1/2 NET and G3 NET [227].

In terms of molecular markers, the negativity of intra-tumor synaptophysin is associated with significantly shorter PFS with sunitinib treatment [227]. Another study involving G3 GEP-NEN patients reported that intra-tumoral expression of phospho-AKT was significantly associated with lower radiological response to sunitinib treatment (*p* = 0.04) as well as with Ki-67 index (*p* = 0.03), while its effect on PFS/OS was not investigated. 

The predictive value of various tissue- and blood-based molecular factors on response to sunitinib treatment have been extensively studied in other malignancies, specifically in renal cell carcinoma, which are well summarized in [257,262].

## 6. Predictive Factors for Response to PRRT in Advanced panNENs

### 6.1. General: Predictive Factors with PRRT

The results of a recent Phase III (NETTER-1) trial demonstrated the significant anti-tumor effect of ^177^ Lu-DOTATATE in patients with advanced SSTR-positive midgut NENs [58,83] (Table 2, Left Panel). The estimated PFS at 20 months in patients treated with ^177^ Lu-DOTATATE was 63%, which was significantly higher than that of patients treated with octreotide (10.8%) [58]. The efficacy of PRRT was also reported from a pivotal retrospective study involving many panNEN patients, with median PFS and OS of 33 months and 46 months, respectively [59], as well as in other studies [263]. Similarly, in a recent systematic analysis of published studies involving 697 patients with advanced panNENs, PRRT treatment was more effective/safe/better tolerated than the use of everolimus [61]. Specifically, PRRT was superior to everolimus treatment in objective tumor response rate (47 vs. 12%, *p* < 0.001); better disease control rate (81 vs. 73%, *p* < 0.001); resulted in a longer PFS (25.7 vs. 14.7 mos., *p* < 0.001); had a better safety profile with less grade 3//4 toxicity (5 vs. 11%, *p* = 0.02) and numerically had less nephrotoxicity (1 vs. 5%, *p* = 0.34) [61]. Furthermore, treatment discontinuation because of adverse events was lower with PRRT (0%) than with everolimus (16%) [61]. However, at this time, no randomized control prospective Phase 3 trial has been performed targeting panNEN patients only. In other studies, the efficacy of PRRT in patients with advanced panNENs is represented by its high cytoreductive effect, with 46% of patients in one report [59] found to show some degree of tumor shrinkage (ORR 29%) with a markedly prolonged time to progression observed (median, 40 months). In a recent meta-analysis of 1920 NEN patients with advanced disease from 18 studies treated with PRRT, 30.6% had a disease response by SWOG criteria (29.1% by RECIST criteria) and the disease control rate was 81.1% by SWOG criteria and 74.1% by RECIST criteria [55].

The major and characteristic AEs related to PRRT which may have a significant impact on its clinical outcomes include myelosuppression as a short-term AE [264,265], as well as the occurrence of renal failure and leukemia/myelodysplastic syndromes as long-term AEs [265,266,267,268]. Despite kidney protection with coadministration of positively charged amino acids, the median decline in creatinine clearance was observed in 7.3% per year in patients treated with ^90^Yttrium (Y) -DOTATOC, which was higher than that observed in patients treated with ^177^Lu-DOTATATE (3.8% per year, *p* = 0.06) [268]. A recent retrospective analysis involving 521 patients with advanced NENs reports that development of therapy-related myeloid neoplasms was infrequent (4.8%) but resulted in fatal outcomes [267].

Although numerous studies report the efficacy of PRRT in panNEN patients [165,266,269], most studies investigating predictive factors on clinical outcome of PRRT included various types of NENs. Therefore, this section covers the results from these recent studies, even though the proportion of panNENs is small. For previous findings regarding predictive factors on PRRT efficacy from older studies, please refer to [165,266,270]. 

### 6.2. Clinical Predictive Factors for PRRT Efficacy 

Several studies have reported that patients with poorer performance status (≥1) had increased risk of disease progression [111,114,133,184,185] and mortality [59,111,114,184,186,187] with PRRT (Table 2, Left Panel). Other prominent clinical prognostic factors include age [111,133,181,182,183,201], gender [182,183] and diabetes [114]. In terms of functional status, a study involving 131 patients with GEP-NENs (eight gastrinomas, 32 NF-panNENs) reported that patients with gastrinomas had higher frequency of remission (*p* = 0.018) [188]. However, time to progression was significantly shorter (*p* = 0.004) than patients with carcinoid and NF-panNENs [188]. Other studies also report a significantly poorer OS in patients with functioning tumors [59,186], suggesting a difference in anti-tumor activity of PRRT regarding functional status of the panNENs. 

Recent studies report that PRRT, in addition to its anti-growth effects on NEN growth, is also highly effective for refractory functional panNEN syndromes, especially VIPoma’s, malignant insulinomas, and carcinoid syndrome [39,160]. Such an approach is rarely needed in the most common, malignant F-NEN syndrome, Zollinger-Ellison syndrome [1,158], because, at present, in contrast to the past [157,271], the acid hypersecretion [272] due to the ectopic secretion of gastrin by the gastrinoma [154], can be controlled by medical therapy (Proton pump inhibitors, histamine H_2_- receptor antagonists) in almost every patient, even those with multiple endocrine neoplasia-type 1 [157,271,273,274,275,276]. This effect of PRRT on functional NEN activity is independent of the antigrowth effect of PRRT [39,160]. At present there are no studies on predictors for which patients with refractory F-NENS will have their hypersecretory state respond to PRRT. 

### 6.3. Laboratory Test/Biomarkers Predictive Factors for PRRT Efficacy

Various biological markers predictive for outcomes of PRRT have been studied. An elevated baseline level of CgA is associated with both decreased post-PRRT PFS [114,184,189,190] and OS [114,184,190,191,192]. A biochemical response in CgA levels is significantly associated with longer post-PRRT survival [184,193], while increases in CgA levels after PRRT correlates with worse OS [191]. However, according to a study comparing the predictive value of the levels of blood NEN transcripts related to the growth-factor signalome and the metabolome with CgA on PRRT efficacy [134], changes in the blood NEN transcript levels more consistently correlated with treatment response than changes in CgA for both responders (*p* = 0.0002) and non-responders (*p* = 0.007). A similar high predictive value for efficacy of PRRT by assessment of such blood NEN transcripts levels was reported in a second study in patients with advanced NENs [133]. Although neither baseline levels of these markers are predictive of clinical outcomes in one of these studies [134], in the other study the predictive utility on PFS was reported for the quotient, calculated by combining values of the assessment of the blood NEN transcript levels and Ki-67 index [133]. In other studies, elevated levels of NSE [186] and pancreastatin [192] were associated with increased mortality. A study involving 55 NEN patients (eight panNENs) reported that a pre-treatment inflammation-based index score, derived from serum C-reactive protein (CRP) and albumin levels, as well as its change through PRRT treatment, is predictive for both PFS and OS post PRRT [195]. In addition, either CRP or albumin alone, as well as blood somatostatin levels, are also predictive of post-PRRT PFS/OS [195]. In a PRRT study involving 149 patients with G3 GEP-NEN (89 panNENs), high pre-treatment levels of either lactate dehydrogenase or alkaline phosphatase were predictive for worse PFS and OS [111]. Higher ORR is also reported in patients with elevated plasma lactate dehydrogenase LDH levels post PRRT (*p* < 0.05) [111].

### 6.4. Imaging Predictive Factors for PRRT Efficacy 

Several studies demonstrated that higher uptake with SSTR-PET is associated with a favorable outcome with PRRT [112,185,186,197,198,199,200,277], whereas ^18^F-FDG PET positivity is associated with shorter PFS [62,112,134,183,198,201,202,203,278] and OS [62,112,183,202,203,204] (Table 2, Left Panel). Until recently, one of the most used methods for predicting the possible effectiveness of PRRT was to compare the degree of radiolabeled SSA uptake in the liver to the tumor (Krenning score) primarily with ^111^indium-pentetreotide scintigraphy [188]. However, the exact value of this approach or parameters of uptake to use may be different with the widespread use of recently developed ^68^Ga-DOTATE PET/CT, particularly with a small lesion measuring <2 cm [279]. This discrepancy in the Krenning score between different imaging modalities can have an impact on the predictive value of assessing SSTR expression on PRRT efficacy. The results from a study including 65 patients with advanced NET G1/2 (16 panNENs) that reported that intra-tumor heterogeneity of SSTR expression was significantly associated with shorter PFS and OS with PRRT [205]. Several textural parameters of SSTR-PET demonstrating heterogeneity are also reported to have predictive value on PFS/OS [185,189,197,206,207,208] (Table 3), suggesting the importance of pre-treatment detailed visual and texture analysis of SSTR-targeted radionuclide imaging in patients undergoing PRRT.

### 6.5. Treatment-Related Predictive Factors for PRRT Efficacy 

In terms of treatment-related factors predicting response to PRRT, numerous studies report that patients who did not respond to PRRT had poorer PFS [184,199,211,212] and OS [59,184,187,193,212,213,214,215] (Table 2, Left Panel). The failure to resect the primary tumor [114,184,213] is also associated with worse outcomes with PRRT. However, the predictive value of primary tumor resection on response to PRRT still remains controversial and needs further validation by prospective studies. Among two widely used radioisotopes for PRRT, ^90^Y and ^177^Lu, ^177^Lu emits a shorter range of beta particles and lower maximum energy, but has a longer half-life than ^90^Y [266,270]. In addition, ^90^Y and ^177^Lu have different abilities to target large and small lesions, respectively, depending on their different penetration range [266,270]. Several studies report the superior anti-tumor activity of the combined application of ^90^Y- and ^177^Lu-based PRRT than with ^90^Y or ^177^Lu solely [114,182,217]. However, currently there are no established criteria for selecting these options [280]. Several studies report a significant correlation of lower cumulative dose with decreased PFS [198,207,216] and OS [62,183,198,206,207]. Similarly, a prospective study involving 200 NENs (48 panNENs) reported that patients whose absorbed dose to the kidney reached more than 23 Gy had significantly longer PFS (*p* < 0.0001) and OS (*p* < 0.0001), as well as a higher rate of objective response (31% vs. 13%, *p* < 0.0001) and biochemical responses (80% vs. 45%, *p* = 0.0011) than those who did not [193]. A significant correlation between the absorbed dose and tumor reduction is also reported from another PRRT study [281]. These results can be partially due to the bias that patients with longer PFS/OS are likely to receive more treatment cycles, resulting in subsequent increases in administered dosage [207]. On the other hand, like everolimus treatment, dose-reduction and schedule adjustment are frequently required in a certain proportion of patients to protect renal and bone marrow function, as well as management of AEs [29,269,282]. Therefore, the prognostic value of these protocol-related factors, types of radioisotope and cumulative dose needs confirmation by future randomized, prospective studies.

Prior chemotherapy is associated with both decreased PFS and OS post PRRT [114,184,198,218,219], whereas prior TACE correlates with a decreased PFS [211]. An increasing number of previous therapies is also associated with worse PFS/OS [182], suggesting the importance of timing to induce PRRT in the treatment sequence in patients with advanced panNENs. In contrast, some recent studies report the combination of chemotherapy and PRRT may led to an increased disease control rate [264,283].

In terms of treatment-related factors predictive for the occurrence of specific AEs, renal radiation dose, in addition to other clinical factors including higher age, presence of hypertension and diabetes, are probable contributing factors to decreased creatinine clearance [268]. The prior number of therapies, prior chemotherapy with alkylating agents and prior radiotherapy, as well as other factors including higher age (>70 years), baseline cytopenias and impaired renal function were associated with PRRT-induced myelotoxicity [265]. 

### 6.6. Pathological Predictive Factors for PRRT Efficacy [Histological Factors/Classification/Grading, Molecular Factors] 

Several studies report that NEN patients with G3 tumors have an increased risk of disease progression [110,114,182,183,218,222,284] and mortality [36,114,134,182,183,214,217,218,222,284] compared to patients with G1/2 tumors post PRRT (Table 2, Left Panel). For well-differentiated tumors, patients with G2 tumors are also reported to be associated with worse PFS [211] and OS [36,114,182] than patients with G1 tumors post PRRT. In terms of Ki-67 index, the usefulness of cut-off values proposed by the 2010 WHO classification system to separate G1/G2 [186,193] and G2/G3 [193] is validated by several studies, while other studies report the better predictive value on PFS with the use of a Ki-67 threshold of 5% [114,186] and on OS with the use of 10% [114] post PRRT, respectively. 

The recent 2017 WHO classification revision dividing G3 panNENs into well differentiated G3NET and poorly differentiated G3NEC has important implications for treatment, including the use of PRRT [110,112,284]. The expression of SSTR2A in G3 NENs is reported from several studies [285,286], even in poorly differentiated tumors [226,285,287,288,289,290], as well as in tumors with a higher Ki-67 value (>50%) [291]. Despite a lower frequency [286,288,290] and weaker expression levels [285] of SSTR2A in G3 tumors than in G1/2 tumors, its expression strongly suggests the potential value of the anti-tumor effect of PRRT in patients especially with G3NETs, and even a proportion with G3NEC [110,111,112,284,292]. Accordingly, the efficacy of PRRT in patients with G3 tumors is reported from different groups [110,111,112,284,292]. A recent study including 149 patients with G3 GEP-NENs (89 panNENs) reported that a Ki-67 index of ≥55% and the presence of a poorly differentiated tumor are independent predictive factors for worse clinical outcomes of PRRT, although these factors did not affect ORR [111]. These results of a lower PRRT response in patients with G3 NEC (poorly differentiated) vs. G3 NET (well-differentiated) could be partially explained by a study demonstrating that panNEN patients with G3 tumors express SSTR2A in 78% of G3 panNETs and only 42% of G3 panNECs [293]. The predictive significance of the Ki-67 threshold of 55% on PRRT efficacy is also reported from a study involving 69 patients with G3 NENs (46 panNENs) [112]. These findings can be important for optimal treatment selection in patients with G3-panNENs as well as other G3-NENs, because of a large proportion with advanced diseases, a significantly short survival time, as well as limited effective therapeutic options for these patients.

Patients with bone metastases [59,183,184,187,188,190,198,214] as well as with higher hepatic tumor load [59,62,183,184,186,190,213] are reported to have a worse clinical outcome of PRRT. In addition to the findings that poor performance status, as well as previous treatment correlate with worse outcomes, as mentioned above, early initiation of PRRT can maximize its anti-tumor activity [282], which needs to be validated by prospective, randomized studies with other therapeutic agents. 

## 7. Predictive Factors for Response to Chemotherapy in Advanced panNENs 

### 7.1. General: Predictive Factors with Chemotherapy

The frequently-used regimens for chemotherapy in advanced panNEN patients included streptozotocin-, temozolomide- and dacarbazine-based regimens for patients with well-differentiated G1/2 tumor and platinum-based regimens with G3 tumors, respectively [65,66,67,68,87,108] (Table 4). A recent multicenter, randomized study [87] [ECOG-ACRIN Cancer Research Group-E2211] compared temozolomide (TEM) alone to TEM plus capecitabine (CAP) [CAPTEM] in 144 patients with advanced, progressive panNENs (Grades G1/G2). The PFS was greater with CAPTEM than TEM alone (22.7 vs. 14.4 mos., HR = 0.41, *p* = 0.023). Mean follow-up was 29 mos., and the median OS was greater with CAPTEM than TEM alone (38 mos. vs. not reached, HR = 0.441, *p* = 0.012) [87]. The treatments were well tolerated with the expected adverse events and with higher rates in the combination arm. 

Recently, the 2017 WHO classification system for panNENs was modified, dividing G3 tumors into G3 NETs and G3 NECs based on their differentiation with G3 NETs being well-differentiated and thus like G1/2, whereas G3 NECs are poorly-differentiated [294]. Not only are these two G3 groups different in molecular determinants/pathogenesis, but they also differ in their anti-tumor treatments and chemotherapeutic approaches [284,295,296,297]. In general, G3 NECs are initially treated with platinum-based chemotherapy [65,66,67,284]. In contrast, the efficacy of platinum-based chemotherapy is reported to be limited in patients with G3 NETs [295,298,299]. If chemotherapy is needed in patients with G3 NETs, alkylating agents (temozolomide- or dacarbazine-based treatment) have been reported to be effective (ORR 50%), which is like the effectiveness of this regimen in G3 NECs (ORR 50%) [65,300]. Therefore, if chemotherapy is considered for G3 NETs, they are treated like G1/2 NETs with temozolomide- or dacarbazine-based chemotherapy, PRRT, or targeted therapy; however, which of these treatments is most efficacious and should be preferred is unclear [301]. In addition, currently there has been a lack of evidence which directly compared the anti-tumor activity within these regimens, as well as with other therapeutic agents. However, the higher cytoreductive effect of chemotherapies than other therapeutic agents may indicate an important role in treatment for the purpose of tumor volume reduction [106].

### 7.2. Clinically-Related Predictive Factors for Chemotherapy Efficacy [Clinical, Laboratory, Imaging, Treatment-Related Factors] 

Poorer performance status is reported to be significantly associated with decreased PFS/OS with chemotherapy in panNEN patients [302,303,304,305,306,307], as well as those with G3 tumors [298,308] (Table 4). Higher age [304,305,309,310,311] and the prior lack of primary tumor resection [302,306,312,313] are also reported to correlate with worse outcomes with chemotherapy, while the effect of functional status remains controversial [302,312]. 

Several studies report the predictive value of tumor biochemical response assessed by changes in CgA levels for PFS [305,306,314] and OS [315] in panNEN patients treated with chemotherapy. Other studies reported that urine positivity of 5-HIAA [305], higher neutrophil-to-lymphocyte ratio, platelet-to-lymphocyte ratio [316], as well as elevated pre-treatment levels of alkaline phosphatase and CRP [317] were associated with decreased PFS with chemotherapy. For panNEN patients with G3 tumors, elevated baseline levels of serum lactate dehydrogenase [298,308] and increased platelet count [298] correlate with worse survival with platinum-based chemotherapy.

In terms of imaging factors in predicting chemotherapy outcomes, a recent study involving 173 advanced NEN patients (79 panNENs) reported a significantly improved OS in patients with somatostatin receptor scintigraphy (SRS)-positive tumors than those with SRS-negative tumors [304]. In addition, another study showed significantly higher ORR in SRS-positive tumors [306]. The predictive value of other various imaging factors on response to chemotherapy have been renewed recently and will not be discussed further here [47]. 

Like the predictive factors for other treatment modalities, as mentioned earlier, patients who achieved radiological response had longer PFS [306,313,314], whereas those who achieved disease stabilization had longer survival [306]. Several studies reported that prior chemotherapy was significantly associated with worse outcomes with regard to another course of other chemotherapies [229,302,303,310,311,317,318].

### 7.3. Pathological Predictive Factors for Chemotherapy Efficacy [Histological Factors/Classification/Grading, Molecular Factors] 

Numerous studies report a correlation between higher tumor grade and worse outcomes with chemotherapy, with most of these studies showing significantly decreased PFS/OS in patients with G3 tumors [108,303,309,312,316,318,319,320,330] (Table 4). Furthermore, numerous studies report the predictive utility of an increasing Ki-67 index correlating with worse PFS [302,303,305,306,310,318,321,322,323] and OS [304,305,310,312,318,321,323,324] with chemotherapy (Table 4). Besides the validity of a Ki-67 cut-off value proposed in the 2010 WHO classification system [304], other studies report the superiority of an even higher Ki-67 threshold in predicting chemotherapy outcomes, including 5% [312], 10% [305,306] and 15% [321]. For patients with G3 tumors, a previous pivotal study including 305 patients with GEP-NECs (71 panNECs) demonstrated that patients with Ki-67 <55% had a significantly lower ORR (*p* < 0.001), but a longer OS (*p* < 0.05) on platinum-based chemotherapy than patients with higher Ki-67 values [298]. This is consistent with another study that analyzed 70 patients G3 panNETs/NECs and reported a Ki-67 value of >55% as an independent predictor for poor prognosis [295] post platinum-based chemotherapy. It is also reported that within G3 NENs, patients with poorly-differentiated tumors had a significantly higher rate of radiological response/stabilization, but a shorter OS with chemotherapy [295,299,307,319,323,324,325,326]. The importance of the differences in differentiation is emphasized in the latest 2017 WHO classification system for panNENs (i.e., G3 NETs vs. G3 NECs) [294], as mentioned above, which will help clinicians to predict responsiveness to platinum-based chemotherapy, specifically in patients with G3 tumors. In a recent meta-analysis, the panNEN grade was superior to T, N, or M status in predicting outcomes and selecting patients for chemotherapy in patients with advanced disease [68]. In this analysis of the NCI SEER data [68] a significant improvement in survival with chemotherapy was only seen in patients with poorly differentiated and undifferentiated panNENs. Patients with extrahepatic metastases are reported to have a significantly worse PFS [302,303], as well as decreased ORR [302] with chemotherapy. Another study involving 143 panNEN patients treated with temozolomide and capecitabine reported that alternative lengthening of telomeres (ALT) positivity was associated with improved OS, while it did not predict response to this regimen [327]. In addition, the expression of DAXX/ATRX was not associated with outcomes of chemotherapy with these agents [327]. According to a study investigating the predictive value of 5-fluorouracil (5-FU)-related factors in 41 patients with panNENs, the intra-tumor expression of thymidylate-synthase was an independent predictor for improved PFS with a 5-FU/streptozotocin regimen [305]. Higher expression levels of dihydropyrimidine-dehydrogenase are also significantly associated with radiological (*p* = 0.018) and biochemical (*p* = 0.04) response to 5-FU/streptozotocin treatment, whereas it had no effect on PFS [305].

One of the most investigated molecular factors predictive for chemotherapy outcomes is promoter methylation status and the expression level of MGMT (O^6^-methylguanine-DNA methyltransferase) [300,318,327,328,329,331,332,333]. However, the predictive value of MGMT status is controversial, showing its importance in some studies [318,328,329,331] but not others [300,327,332,333]. The cytotoxic effect of alkylating agents is due to the alkylation of DNA bases that can impair essential DNA processes such as DNA replication and/or transcription [334]. The O^6^-position of guanine in DNA is the most frequent site of alkylation by these agents which generate O^6^-methylguanine (O^6^meG) [334]. MGMT plays an important role in repairing alkylating-induced DNA damage, and thus the reduction of its activity has been reported to be predictive of improved therapeutic response to chemotherapy using alkylating agents [334,335]. Several recent studies reported that the presence of MGMT promoter methylation [318,328,329] and loss of protein expression [328,329,331] were significantly associated with higher ORR, as well as prolonged PFS/OS post chemotherapy in NEN patients. A recent meta-analysis of the effect of MGMT status on response to TEM in advanced NENs was reported [336]. In 513 NEN patients from 12 studies the ORR was higher in MGMT-negative patients with a risk difference of 0.31, *p* < 0.001, risk ratio of 2.29, *p* < 0.001, and a pooled PFS hazard ratio, HR = 0.56, *p* < 0.001 and OS HR-0.41, *p* = 0.011, comparing MGMT-deficient to MGMT-proficient NENs [336]. In contrast, several other studies also report that the assessment of MGMT status was not informative to predict clinical outcomes of chemotherapy [300,327,332,333]. Therefore, together with the conflicting results from older studies [335], the predictive utility of MGMT status on outcomes with chemotherapy remains controversial and is not universally assessed in panNENs. To address this issue, the predictive value of MGMT status on response to alkylating agents in patients with NENs is now under investigation as a primary endpoint in a randomized, prospective study comparing alkylating-based and oxaliplatin-based chemotherapy (NCT03217097) [335], and as a secondary endpoint in a Phase 2 study with lanreotide plus temozolomide (NCT02698410), as well as in a randomized Phase 2 study comparing temozolomide alone or in combination with capecitabine (NCT01824875). 

A study investigating predictive molecular markers for response to platinum-based chemotherapy in panNEN patients with G3 tumors reported that a KRAS mutation and/or loss of Rb expression were predictive for worse clinical outcomes post platinum-based regimens [295]. The predictive value of Rb loss remained statistically significant among patients with poorly-differentiated G3 panNEC [295], suggesting its additional prognostic information among patients with these types of highly-aggressive tumors.

## 8. Predictive Factors for Response to Liver-Directed Therapies in Advanced panNENs

### 8.1. General: Predictive Factors with Liver-Directed Therapies

The importance of neuroendocrine liver metastases (NELM) in the survival of patients with advanced panNENs is represented not only by the fact that presence of liver metastases is associated with worse survival [11,70,155,162,337,338,339,340,341], but also by a significant correlation between the higher tumor burden of NELM and worse outcomes with PRRT [59,62,186,213], SSA [173,174,177,179,225], liver-directed therapy [342], and chemotherapy [321] (Table 5). Several studies have reported the efficacy of liver-directed therapies including TACE/TAE and radio-embolization using ^90^Y-labeled microspheres (selective internal radiotherapy [SIRT]), in controlling focal progression of NELM [69,70,71,72,74,341,343], as well as in controlling symptoms due to the liver metastases or hormone excess state of a F-panNENs [9,39,70,71,74,341,344]. The mean overall ORR to radio-embolization in a review of 12 studies including >400 patients with unresectable NELM was 55% (range 12–89%) and stable disease in 32% (range 10–60%) [9]. In a recent systematic analysis/meta-analysis of radioembolization/SIRTS for NELM (27 studies) [345] the pooled estimate of the ORR was 51% (95% CI-47–54%) and the disease control rate was 88% (95% CI-85–90%) with a median overall survival post SIRT of 32 mos. 

In several studies with TACE/TAE in patients with advanced NENs, the ORR was 25–85% and the symptomatic response was 50–100% [9,70,71,74]. In a recent systematic review [74] of embolization studies for neuroendocrine liver metastases (NELM) (101 studies, 5545 patients), the pooled partial response rate was 36.6%, 38.9% for stable disease, and 55.2% had a symptomatic response to treatment. The mean PFS and OS was 18.4 mos. (95% CI-15.5–21.2 mos.) and 40.7 mos. (95% CI-35.2–46.2) [74]. Although there is only limited data, in panNENs TACE appears to be more effective than TAE, whereas they have similar effectiveness in patients with advanced NELM from carcinoids/non-panNENs [346].

TACE/TAE requires multiple sessions, whereas radio-embolization provides a similar anti-tumor effect with a fewer (usually single) number of sessions and less toxicity compared to TACE/TAE [72]. However, because of a lack of randomized, prospective studies including a large number of patients, as well as a lack of studies comparing different treatment modalities in a homogenous group of patients, the current position of these liver-directed therapies in the multimodal approach to NELM treatment remains unclear. 

Because there is no study reporting predictive factors on outcomes with liver-directed therapies only in panNEN patients, below we include results from the studies including various types of NENs, even if the proportion of panNENs is small.

### 8.2. Clinically-Related Predictive Factors Liver-Directed Therapies Efficacy [Clinical, Laboratory, Imaging, Treatment-Related Factors] 

The clinical predictive factors for poor response with liver-directed therapy’s efficacy include higher age [347,348] and poorer performance status (≥1) [349,350,351]. In terms of biomarkers, elevated pre-treatment levels of serum CgA [347], pancreastatin [342,352], bilirubin [353], alkaline phosphatase [354,355], and Child-Pugh score [351,355] are associated with worse OS post liver-directed therapies (Table 5). In another study, a worse PFS/OS was reported in patients with increased CgA levels after TACE [356]. 

A study involving 51 patients with NENs (21 panNENs) reported that the occurrence of an early response (≥50% decrease) showing a decreased tumor burden assessed by MRI imaging was an independent predictor of worse OS post TACE [350]. The predictive utility of assessing lung-shunt fraction on radio-embolization efficacy is reported from a study including 44 NEN patients (17 panNENs) [353].

Like the other treatment modalities, patients who experienced a radiological response to liver-directed therapies had better OS than non-responders [347,348,351,357]. It is also reported that an increasing number of TACE sessions is associated with better survival [347,352], which may be the case since patients with longer PFS are more likely to receive many sessions of TACE. The reason for a significant association between the presence of enterobiliary communication and a worse OS with liver-directed therapies can be explained by the poor survival rates in patients undergoing pancreaticoduodenectomy or common bile duct stent insertion caused by a mass located in the pancreatic head or lymph node metastases [358]. Several studies report that patients who received prior systemic therapy had worse outcomes post liver-directed therapies [348,353,355,358,359]. The concomitant use of SSA correlated with improved OS [348,356], whereas systemic therapy during the follow-up period was associated with worse PFS [349]. In one Phase II study [360] involving 39 patients with NELM, following TAE, they were treated with sunitinib, and a high ORR of 62%, a median PFS of 15.2 mos. and an overall four-year survival of 59% (95% CI-38–80%) was reported. The true prognostic value of these treatment-related factors with liver-directed therapies remains unclear and needs to be validated by future prospective studies.

### 8.3. Pathological Predictive Factors for Liver-Directed Therapies Efficacy [Histological Factors/Classification/Grading, Molecular Factors]

Several studies showed that patients with higher tumor grade had worse PFS [348,349,356,359] and OS [342,347,348,349,351,354,356,357] with liver-directed therapies, particularly in patients with G3 tumors [342,347,349,357]. Pathologic factors predicting worse outcomes with liver-directed therapies include high hepatic tumor burden [342,347,348,349,352,354,357,358,361,365] (Table 5). The most frequently reported cut-off value of hepatic tumor burden predictive for worse survival post liver-directed therapies is 75% [342,347,352,357], followed by 50% [342,348,349,354,365], 25% [342,365], 20% [358] and 10% [365]. Several studies reported that NEN patients presenting with extrahepatic metastases [347,348,350,355,357,358,361,362,365], ascites [355] or portal vein thrombosis [350] had a significant decreased PFS/OS with liver-directed therapies. These results suggest an increasingly limited role of liver-directed therapies options in NEN patients presenting with extrahepatic lesions. In the past, one of the main indications for liver-directed therapies in such patients was the presence of unresectable tumors causing a functioning tumor syndrome which was difficult to control [39,274]. However, with the increasing effectiveness of other less invasive therapies, such as molecular targeted therapies, SSA and PRRT, this is less of an indication for liver-directed therapies [39,274]. Because of the lack of well-established indication criteria for TACE, TAE and radio-embolization in panNEN patients, these histological factors may provide important clinical information in deciding on therapeutic strategies, specifically for patients whose tumor location is limited to the liver.

In terms of molecular predictive factors for the efficacy of liver-directed therapies, a recent study involving 51 patients with NENs (23 panNENs) reported that mutations in DAXX (*p* < 0.001) or MEN1 gene (*p* = 0.018) are significantly associated with shorter hepatic PFS after TAE. The predictive value of DAXX mutation on hepatic PFS post liver-directed therapies is also statistically significant when the results are analyzed only in patients with panNENs (*p* = 0.026) [359]. DAXX is reported to activate JNK-mediated apoptosis under hypoxic condition, and thus its loss disrupts ischemia-induced cell apoptosis [366], which can result in sustained tumor growth after TAE. These data suggest that identifying the mutation-status of DAXX, as well as other DAXX-related factors, such as ATRX and ALT [297,367], can provide important prognostic information in panNEN patients undergoing liver-directed therapies; however, before routine use can be recommended, additional confirmation by future prospective studies is needed.

## 9. Controversies and Uncertainties of Predicting Therapeutic Response in Advanced panNENs

Despite the increasing number of recent insights regarding the factors predictive for therapeutic responses to each treatment option, as reviewed in the previous sections, several controversies and uncertainties remained in these areas (Table 1).

For several reasons, one of the most prominent controversies/uncertainties is the lack of well-established evidence for the routine clinical use of the various proposed prognostic factors predictive for optimal treatment selection in the treatment cascades in advanced panNENs. This has occurred for a number of reasons. First, because of the recent increase in several available therapeutic options, markers to predict which patient will benefit from which treatment option are highly warranted. This has occurred because of the lack of a high level of evidence which has prospectively compared the anti-tumor effect of these therapeutic agents with randomization in various cohorts of patients, and thus factors predicting therapeutic response are becoming increasingly important. Each of these different therapeutic options have similar indication criteria (e.g., age > 20, G1/2 NET vs. G3 NEC, preserved organ function and performance status, had an absence of severe complications, etc.) which fits all of the patients, although the disease course and malignant behavior vary markedly between individual patients. Furthermore, several common prognostic factors, such as age, performance status, CgA, grade and disease extent may be important in predicting each treatment response, but not be useful in treatment selection among different options. Therefore, the establishment of selection criteria specific for each of these therapeutic options is needed. 

Second, as many patients with advanced panNENs are now increasingly receiving multiple lines of treatment, predictive factors to tailor the order of the different treatments in the therapeutic cascades is becoming increasingly important. Although several guidelines proposed from various societies [5,166,167,168,169] have suggested orders of the different treatments in patients with advanced panNENs, there are few prospective studies to support these recommendations. Furthermore, when therapies are similar in efficacy/AEs, there is even less data to support any proposed order of treatment, such as treatment with everolimus vs. sunitinib in various groups of advanced panNENs. Whether prior treatment affects the efficacy and toxicity of the following treatment can have a significant impact not only on the therapeutic response of each single treatment. This needs to be carefully investigated prospectively in future systemic studies.

The order of administering the different anti-tumor treatment modalities is also strongly affected by their different degrees of anti-tumor activity as well as toxicity profiles. The optimal treatment selection can be strongly affected by whether the goal of the treatment is to achieve long-term disease stabilization or to reduce tumor burden. In general, the former may be a good indication for tumoristatic agents (SSAs, everolimus and sunitinib), whereas for tumor reduction, tumoricidal agents, such as PRRT, chemotherapy and liver-directed therapies will be preferred. Treatment decision making is also based on the toxicity and tolerability of each therapeutic agent, and thus factors predictive for objective response, as well as the occurrence of specific AEs, are becoming increasingly important, together with better understanding of the characteristics of each therapeutic option. This can also be important in another controversial area which deals with which treatment could be suitable for neoadjuvant/adjuvant therapy in panNEN patients [7]. There is little data in this area, which means that controversy exists, not only with regard to what treatment might be effective, but when, in whom, and for how long it should be used. In this regard, treatments with both high cytoreductive effect and low rate of disease progression are needed for neoadjuvant therapy to reduce tumor size (i.e., downstaging), as well as for adjuvant therapy to prevent disease recurrence in a high-risk group [7]. Recently, PRRT is receiving increasing attention as a possible neoadjuvant therapy in panNENs with its high rate of radiological response [181,368,369,370,371], while evidence to support its applicability in the real clinical practice still needs to be validated prospectively. 

Third, factors to predict the appropriate timing to stop or switch the treatment have become increasingly important in the aspect of treatment sequence, and because of the lack of systematic studies, there is no uniform agreement. In terms of long-term outcomes with multimodal treatments in patients with panNENs, whether each of the treatments should be continued until disease progression is still unclear. Recent studies report the usefulness of tumor growth rate calculated at an early phase (3 months) after treatment initiation in predicting PFS and radiological response [372,373], which may provide additional information in optimizing the treatment duration of each therapeutic option. In addition, there has been no standardized criteria with regard to how many sessions of liver-directed therapies ought to be performed, specifically with TACE and TAE, to maximize the anti-tumor effect without increasing the occurrence of AEs.

Lastly, whether combination therapy of these therapeutic agents will provide additional clinical benefit remains controversial. As mentioned earlier, combination therapy with everolimus and other agents has been examined in several previous clinical trials, all of which failed to show their synergistic anti-tumor effects [29]. However, several systemic studies and case series showed promise in combining PRRT and other treatment options, particularly chemotherapy with capecitabine, for both their additional anti-tumor effect and as a radiosensitizing agent [264,266,270]. Given the increasing importance of PRRT in the multidisciplinary treatment in advanced panNENs, further studies to strengthen its clinical use are highly warranted.

Other major controversies/uncertainties for detecting resistance and predicting response/survival/prognosis with various treatment of advanced panNENs includes the role of biomarkers, imaging modalities and molecular markers. In terms of biomarkers, CgA continues to be widely used for NEN diagnosis, as well as a monitoring tool for effectiveness of different anti-tumor treatments in patients with panNENs; however, its serum levels are frequently affected by several factors, particularly using proton pump inhibitors, which are now used for various indications in many patients worldwide [139,142]. Numerous studies demonstrated that in general, serum CgA level/changes have relatively low sensitivity and specificity [139]. Furthermore, its widespread utility is limited by the lack of general agreement on which CgA assay should be used. Recently, the NETest or other related tests assessing blood NEN transcripts are receiving increasing attention as novel blood biomarkers demonstrating higher sensitivity and specificity than other existing diagnostic modalities [131,132,137,374,375]. However, a recent study reported that the NETest has lower specificity than CgA, and the authors concluded that this could limit its diagnostic usefulness, but because of its high sensitivity, they postulated that it may have a role in predicting treatment responses [130]. Recently, a study reported, in the case of PRRT, the predictive value of levels of blood NEN transcripts related to the growth-factor signalome and the metabolome compared to changes in blood CgA for predicting PRRT efficacy [134] and changes in the blood NEN transcript levels more consistently correlated with treatment response than changes in serum CgA for both responders (*p* = 0.0002) and non-responders (*p* = 0.007). At present, the NETest or assessment of other related tests involving the determination of blood NEN transcript levels is not routinely used for either diagnosis or assessing response to treatment and has not been recommended by either ENETs or NANETs for routine use. 

Recently developed imaging modalities, particularly molecular imaging studies, are becoming increasingly important not only for their ability to establish disease localization/extent, but also in providing important prognostic information in patients with advanced panNENs/NENs [47,48]. However, there is no standardized protocol about the follow-up schedule, or in terms of which modalities are to be used in each individual patient and, therefore, they are not widely or routinely used in a uniform manner. 

Several studies have investigated the prognostic utility of various molecular markers in predicting therapeutic outcomes of nonsurgical treatments. Some of those demonstrated promising results, such as the assessment of SNPs of VEGFR with sunitinib (Table 3) and the assessment of the mutational status of MGMT, KRAS, and Rb in chemotherapy (Table 4); however, their prognostic utility has only been shown in a limited number of patients in limited situations. For routine use, when or in which patients these should be used is not clear at present. All of these controversial/uncertain aspects in these areas need to be validated in the future prospective studies, including various cohorts of patients in various situations to support their clinical routine use.

There are also numerous controversial/uncertain areas remaining with the use of each of the specific therapeutic modalities, which have led to the increased importance of prognostic factors in predicting response/survival with each treatment. In the case of tumoristatic agents (everolimus, sunitinib and SSAs), their direct effect on overall survival/prognosis has not been established in the existing Phase 3 trials, primarily because of the use of PFS as a primary endpoint, and because of cross-over after the study period [23,24,46]. The similarities in the anti-tumor effect and toxicities between these agents, particularly between everolimus and sunitinib, have led to queries about which agents are to be used in which patients, as well as the appropriate order for sequencing these agents, due to the lack of head-to-head comparison of the survival benefit between these therapeutic agents. In addition, factors predictive for acquired resistance, as well as for the occurrence of short- and long-term AEs, might be important in treatment selection, which also remains unclear.

There are several controversies/uncertainties remaining with PRRT, including appropriate protocol/dosage/schedules in balancing efficacy and toxicity, factors predicting the occurrence of severe AEs, the efficacy of combination/maintenance therapy, as well as its potential as an adjuvant/neoadjuvant therapy. Dose-reduction and schedule adjustment are frequently required to manage PRRT-related AEs [29,269,282], while an accumulative dose correlates with better therapeutic outcomes of PRRT [62,193,206,207,216]. Therefore, standard methods for routine use which allow appropriate dose adjustment need to be established. The occurrence of specific AEs, particularly myelotoxicity, can result in fatal outcomes from PRRT, as well as preventing success from subsequent treatment [265,268]. Predictive factors to identify what patients will be most prone to develop these complications are essential. With increasing efforts to increase the proportion of patients who respond to PRRT with the use of radio-sensitizers or agents to increase somatostatin receptor expression, this issue is of even greater importance. Several studies have suggested the synergistic anti-tumor effect of combining PRRT [376,377] with other treatments. SSAs [220], everolimus [378] and chemotherapy with capecitabine and temozolomide, can function as radio-sensitizer [379] with PRRT. Furthermore, hepatic radio-embolization with ^166^Holmium [380], and the use of histone deacetylase inhibitors can increase effectiveness of PRRT [381,382]. However, the effectiveness of these combination therapies need to be validated in future randomized, prospective studies, as well as in the careful assessment of their toxicities. The high cytoreductive and tumoricidal effects of PRRT are attracting increasing attention, particularly as a possible neoadjuvant agent in locally-advanced panNENs, while its clinical applicability has not yet been established. 

Despite the high anti-tumor activity of chemotherapy in advanced panNENs, its current position in the multimodal treatment cascade has not been established, primarily due to the lack of significant evidence from systemic prospective studies, as well as from comparative treatment modality evaluation. In patients with well-differentiated G1/2 tumors, it is still unclear which patients will benefit from chemotherapy, the appropriate timing to begin the therapy, or, if it has an effect, when to stop it or restart it. Although platinum-based chemotherapy is widely accepted as a standard initial treatment in patients with poorly-differentiated G3 panNECs, evidence for the best effective subsequent treatment regimens is still lacking. Furthermore, currently there is no established treatment algorithm for patients with well-differentiated G3 NET tumors.

In general, the role for liver-directed therapies is unclear, as is when it is to be used and which liver-directed procedure should be used. Furthermore, its overall role is decreasing in that one of the main indications was for uncontrolled F-panNENs/NENs syndrome, which is now generally well controlled by the other less-invasive therapeutic agents. There is a lack of prospective, randomized studies establishing the exact role of liver-directed therapies and establishing its place in the therapeutic cascade in patients with advanced panNENs. In addition, which patient is the best candidate for TACE/TAE or radio-embolization (SIRT) is currently uncertain. Therefore, sensitive factors in predicting their therapeutic response, as well as disease progression, are highly required to establish their position in the multidisciplinary therapeutic approach in patients with advanced panNENs, in addition to the effective maintenance therapy to delay disease progression post liver-directed therapies. Furthermore, the use of radio-embolization using ^90^Y-labeled microspheres is relatively new, and besides establishing its advantages compared to TACE/TAE, its long-term toxicity needs to be clearly defined.

## 10. Conclusions

This paper reviews in detail the recent progress, as well as the current situation and remaining controversies/uncertainties in attempting to identify predictive/prognostic factors for identification of resistance manifested by an decreasing effectiveness of each of the available nonsurgical therapeutic agents used in the treatment of patients with advanced panNENs. This review demonstrates that several studies have reported the potential predictive value of numerous clinically-related and pathological factors on outcomes with each of these therapeutic options. Prominent clinically-related factors include age, performance status, serum CgA and the presence of prior systemic therapies, as well as prominent pathological/tumoral molecular factors including grade, Ki-67 index, disease extent, and various molecular/genetic tumoral changes which showed significant predictive value for various treatments, often without specificity for one treatment option. These may be useful in predicting therapeutic response, but are not helpful in selecting a particular treatment or in tailoring sequential treatments. Some studies show a possible promise for the specific predictive value on efficacy of a single treatment, such as the occurrence of several AEs with everolimus, the assessment of VEGFR status with sunitinib, SSTR-PET imaging findings with PRRT, and as MGMT status with chemotherapy. Recent studies report promising results with the assessment of a specific subset of the NETest blood transcript levels in PPRT, but it is unclear at present whether a similar approach will be effective with other treatment modalities or if such an approach can be treatment subtype specific. However, at present, none of these predictive factors have been routinely used. Further evidence from prospective, systemic studies are required to support their clinical routine use.

In the last decade there has been a steadily increasing need for the identification of predictive/prognostic factors in all stages of the management of panNEN/NEN patients with advanced disease [7,383,384,385]. This has occurred for a number of reasons, but is largely related to the increasing complexity of treating these patients. This increased complexity is due to the increased availability of different treatment options; the increasing understanding of the natural history of the disease; the increasing understanding of the marked variability of the disease course in different patients which requires different approaches; and lastly, to the increased information available to the physician that could affect management, but at present, often in ways that have not yet been well defined. The status of the development of prognostic/predictive factors in the overall assessment of a panNEN/NEN patient and in the post-surgical management has been dealt with in a recent, separate paper [386]. The present paper’s purpose is to provide a review/analysis of the status/advances/controversies in the identification of predictive/prognostic markers that could be clinically useful in the different nonsurgical anti-tumor therapeutic approaches that are now being increasingly used in these patients. 

One aspect contributing to the increasing need for treatment specific prognostic markers is the rapidly increasing complexity in the nonsurgical treatment of panNEN patients with advanced disease. In the past, the therapeutic approaches were primarily limited to surgical resection in a small percentage of panNEN patients with advanced disease (usually <15–20% of patients), liver-directed therapies consisting of TACE/TAE/RFA, and chemotherapy [9,387]. The recent approval of everolimus [23,29], sunitinib [24], lanreotide [46], as well as PRRT [39,59], and the increasing use of newer liver-directed therapies such as radio-embolization (SIRTS) [388,389] and other somatostatin analogues such as octreotide-LAR, have both expanded the treatment repertoire, as well as increased its complexity [5,9,17,29]. 

The complexity for the clinician is further compounded by the profound advances in NEN/panNEN imaging [47,48,113], as well as in the development of widely used tumor staging/classification/grading systems [7,390]. Unique to NENs is the development of imaging studies using the frequent over-/ectopic- expression of somatostatin receptors by these tumors to image them, which not only is providing enhanced ability to localize small primaries, but also to better staging of the disease by determining the extent/location of the advanced disease with greater sensitivity and high specificity [47,48,113]. This enhanced sensitivity from somatostatin receptor imaging (SRI) is complemented by the increasing improved in sensitivity/specificity of cross-sectional imaging modalities [CT, MRI], the ability to combine these results with SRI findings with hybrid scanners, and the improvement/definition of the value of other molecular imaging modalities such as ^18^F-FDG PET/CT scanning [47,48,113,125,278,391]. For example, regarding the latter point, ^18^F-FDG PET/CT scanning was until recently generally thought to be of little value in patients with panNETs/NENs because these tumors generally have low proliferative rates. However, an increasing number of recent studies are reporting the enhanced uptake in a subset of panNETs/NENs, which is correlating with increased grades [125,278,391]. It is becoming generally established that panNETs/NENs show heterogeneity [205,206,392], and thus the most advanced grade may not be reflected in the biopsies [125]. For this reason and because of the ability of the ^18^F-FDG PET/CT scanning to identify patients with advanced grades or aggressive tumors, it is increasingly being recommended that ^18^F-FDG PET/CT scanning be more widely used, if not used routinely [48,125,278,391,393]. 

Recently, independent of the improved prognostic/predictive results from the enhanced tumor localization information provided by SRI/other imaging studies, there has been substantial advances in the development of increasingly sensitive, radiological prognostic factors generated from enhanced analysis of the image itself provided by each of the imaging modalities, including cross-sectional imaging, SRI and other molecular imaging methods [47,48,123,124,125,278]. Such analyses include texture analyses of the imaging results and the generation of various parameters from analyzing image tumor contrast patterns/kinetics with the computation of imaging modalities such as SUV/Max, etc. [47,48,126,127]. Although at present none of these calculated imaging prognostic parameters are routinely used, there has been remarkable progress made in their predictiveness and potential application. Besides the clear value of possibly predicting the future response/behavior of the panNET with a given therapy, the potential to accurately predict the tumor grade by imaging studies could have a profound effect on the management of a patient. Not only could it reduce the number of biopsies initially needed, but even more importantly, when tumor resistance or growth develops on a given therapy, the imaging results could potentially replace the need for additional tumor biopsies, and even provide insights for alternative approaches if predictive values for tumor specific therapy are developed. 

The development of tumor staging/classification/grading systems by the World Health Organization (WHO), European Neuroendocrine Tumor Network (ENETs) and from the International Union for Cancer Control/American Joint Cancer Committee (UICC/AJCC), have well-established, validated prognostic value [390,394]. These tumor staging/classification/grading systems also identify in some patients specific tumor subtypes (G3-NECs) that may benefit from specific therapeutic approaches and, thus, are of immense value to the management of panNEN/NEN patients [390,394,395]. However, the results of the tumor staging/classification/grading systems can also add an increased level of complexity because, at present, with most patients (80–95%) with an advanced panNENs/NENS having either a G1, G2, or G3 well-differentiated NEN, which may have different prognoses, the best initial treatment to use is not defined in an individual patient, nor is it clear, if tumor resistance or tumor growth develops on a given treatment, what the next treatment should be.

Additional areas leading to the increased complexity for the clinician in managing patients with advanced panNENs/NEN’s are the role of biomarkers, as well as the role of an increasing number of molecular characteristics of the tumors being described that have prognostic value [396]. All would agree that biomarkers potentially could be of great value not only for identifying patients with panNEN/NENs, but also, both for predicting the general prognosis of patients at a given tumor stage. In addition, biomarkers could be of marked value for prediction of responses to specific anti-tumor therapies, tumor recurrence, in addition to identifying early those patients developing tumor resistance to a given treatment [7,396]. While there are an increasing number of reports on the limited role of most widely used biomarkers, such as the assessment of serum CgA levels [140,142], as well as increased reports of newer tumor marketers such as the assessment of circulated gene transcripts [NETest, subsets of NETest] [130,132], or circulating miRNA/lncRNAs/circulating tumor cells, etc. [397,398,399,400,401], at present none are routinely used and it generally remains unclear when these should be routinely used [136]. Regarding molecular advances, there are an increasing number of histological prognosis factors identified in panNETs [DAXX, ATRX, etc.] [367,402,403,404], but at present there are not routinely used.

In each of the above areas of increased complexity for the clinician there have been several advances recently that are providing increasing insights into possible prognostic factors/biomarkers that would have clinical value in nonsurgical antitumor treatment approaches. In this paper these advances as well as the evidence for their current promise for a possible role in patient care is reviewed.

## Figures and Tables

**Figure 1 cancers-14-01250-f001:**
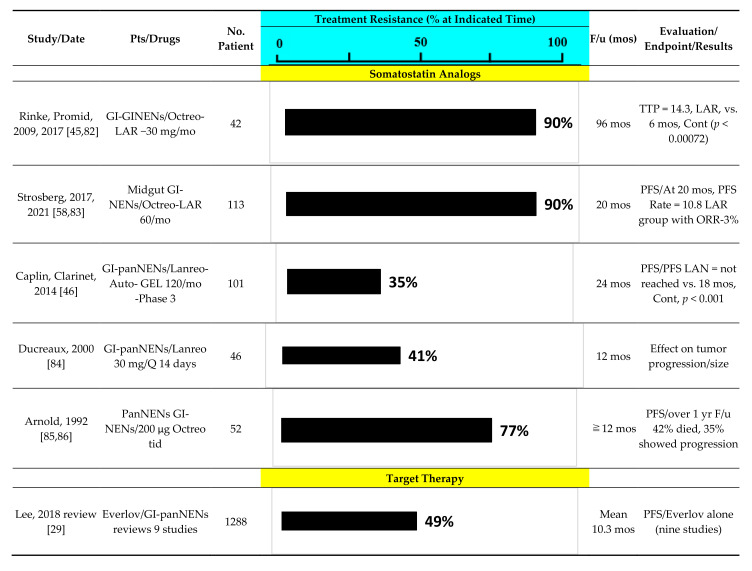
Resistance to various anti-tumor treatments of patients with advanced NENs from various selected studies [24,29,34,35,45,46,58,59,60,62,63,82,83,84,85,86,87,88,89]. This figure only includes data from well differentiated Grade 1 or 2 NENs and includes the percentage of patients that show progressive advanced disease at the indicated times after treatment with the most used anti-tumor agents. Randomized, controlled studies were included whenever available, as well as recent meta-analyses/reviews and summary studies. References for single studies are included in the figure. For the targeted therapy, the data with everolimus are from a review [29] which summarized results from nine studies with everolimus alone [90,91,92,93,94,95,96,97,98,99,100,101]. The data in this figure demonstrated the widespread problem of resistance to the different anti-tumor therapies that are generally used to treat patients with advanced neuroendocrine tumors. Blue and yellow color for emphasis. Abbreviations: TTP—Time to progression; Cont—control; mo—months; GI-NENs—Gastrointestinal neuroendocrine neoplasm; LAR—Octreotide-long-acting release; PFS—progression free survival; ORR—overall response rate; Lanreo—lanrotide; yr—year; F/u—follow up; Everlov—everolimus; SUN—Sunitinib; SURF—Surufatinib; PRRT—Peptide receptor radionuclide therapy; DCR—disease control rate; OS—overall survival; TEM—temozolomide; CAPTEM—capecitabine- temozolomide; BEVA—bevacizumab.

**Table 1 cancers-14-01250-t001:** Reasons for the emerging need for prognostic markers predictive for response/survival and/or development of resistance with specific nonsurgical treatments in patients with advanced panNENs.

**1. Which patients will benefit from each of the available therapeutic options?** - Recent increase in the number of available therapeutic options. - Presence of primary/acquired resistance. - Lack of treatment-specific selection criteria; similar indication criteria between different therapeutic agents.
**2. What is the exact order of various therapeutic options for each individual patient?** - Uncertainty of current position/role of each therapeutic option in the treatment cascade of advanced panNENs. - Does prior treatment affect the efficacy/toxicity of sequential therapies?
**3. Difference in anti-tumor activity/toxicity between different therapeutic agents.** - High cytotoxic/cytoreductive effect to reduce tumor burden? - Long-term disease stabilization with lower toxicity? - Patient tolerance, comorbidities, and toxicity profiles. - How to manage local progression in locally-advanced/metastatic tumors? - Which treatment will be the most appropriate neoadjuvant/adjuvant therapy for patients with panNENs? - Which patients should neoadjuvant/adjuvant therapy be used in?
**4. When to stop or alter the treatment?** - When is the best timing to switch or stop the treatment? - Presence of acquired resistance. - How many cycles/sessions of liver-directed therapies (and PRRT? chemotherapy?) to perform before change/reassessment?
**5. How to follow-up during/after treatment?** - Recent advances in imaging/diagnostic modalities; Which modalities to be used? How often to be performed? - Same follow-up schedule for all?
**6. Which patient will benefit from combination therapies?** - Is there a synergistic anti-tumor effect combining/sequencing these therapeutic agents? - Is the addition of a radio-enhancer effective with PRRT? - Is maintenance therapy needed after PRRT or liver-directed therapies?

**Table 2 cancers-14-01250-t002:** Factors associated with worse PFS and OS with SSTR-targeting therapy (ss and PRRT) in advanced panNENs.

Factors	PRRT		SSA	
	PFS	OS	PFS	OS
**Clinical factor**				
Age	**[63,111]**/[133,181]	**[63,182]**/[183]		[175]
Gender, male		[182,183]	[175]	
Performance status	**[111]**/[114,133,184,185]	**[59,111,114,184,186]**/[187]	** [172] **	
Functioning tumor	** [188] **	**[59]**/[186]		
Symptomatic	** [188] **	**[59]**/[186]	** [173] **	
Progression at baseline	[184]		** [42,173,174] **	
Body weight loss		** [59] **		
Diabetes	[114]	** [114] **		
**Laboratory test/Biological marker**				
CgA, high	**[114,184]**/[189,190]	**[114]**/[184,190,191,192]	**[174]**/[172,175]	[175]
CgA response, no	[184]	[184,191,193]	[176,177]	
NSE, high		** [186] **		
Pancreastatin, high		[192]		
5-HIAA, non-responder			[176]	
Quotient, NETest and Ki-67	[133]			
LDH, high	** [111] **	** [111] **		
ALP, high	** [111,184] **	**[111,184]**/[187,194]	**[173]**/[172]	
CRP, high	**[189]**/[195]			
Somatostatin, high	[195]	[195]		
Albumin, low	[195]	[195]		
Inflammation-based index score	** [195] **	[195]		
WBC, high	** [189] **			
Abnormal blood count		[114]		
NLR, high			** [173] **	
PLR high		[196]		
**Imaging factor**				
SUV, low (SSTR-PET)	**[197,198]**/[112,185,199,200]	[112,186,198]		
SUV, high (FDG-PET)	**[62,183,198]**/[112,134,201,202,203]	**[62,183]**/[112,202,203,204]		
SSTR heterogeneity	** [197,205] **	** [205] **		
SSTR-PET textural parameters	**[185,189,197]**/[206,207,208]	[207]		
Tumor growth rate, high	[209]		[210]	
**Treatment-related factor**				
Non-responder/shorter PFS	**[211]**/[184,199,212]	**[59]**/ [184,187,193,212,213,214,215]		[175]
Lower cumulative dose/reduced dose	**[207]**/[198,216]	**[62,183,207]**/[198,206]		
Absorbed dose to the kidney, <23 Gy	[193]	[193]		
Isotope, ^90^Y/^117^Lu alone	**[182]**/[114,217]	**[182]**/[217]		
No primary tumor resection	[184,213]	**[184]**/[114,213]	** [172] **	
Prior systemic therapy	**[114,184,198,218]**/[182,219]	** [114,182,184,218] **	** [179,180] **	[179]
Prior TACE	** [211] **			
No SSA use (combination/maintenance)	[220]	[220]		
Concomitant use of prior-refracted SSA	[221]	[221]		
**Histological factor/classification/grading**				
Grade	**[182,183,195,211,218]**/ [114,222]	**[134,182,183,218]**/ [36,114,214,217,222]	**[223]**/[42,172]	[179]
Ki-67	**[114,209]**/[111,112,186,193,201,206]	**[114,186,213]**/[111,112,201]	**[172,173]**/[224]	
Differentiation, poorly	[111]	** [111] **	[175]	[175]
Distant metastasis	** [211] **		**[173]**/[224]	
Liver metastasis	[63]	[63,187]	[42]	
Bone metastasis	** [183,184,188] **	**[59,190,198]**/ [183,184,187,214]	** [173] **	
Disease extent	[62]	[187]		
Hepatic tumor burden	[184]/[62,183]	**[59,183,184,186]**/[62,190,213]	**[173,174,179,223]**/ [177,225]	
Ascites	[190]	[190]		
**Molecular factor**				
SSTR2 low		[36]		

Results of the multi-variate analysis are shown in red and bold square brackets to the left of /, while results of univariate analysis are shown to the right of. ALP, alkaline phosphatase; CgA, chromogranin A; CRP, C-reactive protein; FDG, fluorodeoxyglucose; HIAA, hydroxyindoleacetic acid; LDH, lactate dehydrogenase; NLR, neutrophil-to-lymphocyte ratio; NSE, neuron-specific enolase; OS, overall survival; PET, positron emission tomography; PFS, progression-free survival; SSA, somatostatin analogue; SSTR, somatostatin receptor; SUV, standardized uptake value; TACE, transarterial chemoembolization; ^90^Y, ^90^Yttrium; ^117^Lu, ^177^Lutetium.

**Table 3 cancers-14-01250-t003:** Factors associated with worse PFS and OS with targeted therapy in advanced panNENs.

Factors	Everolimus		Sunitinib	
	**PFS**	**OS**	**PFS**	**OS**
**Clinical factor**				
Age			[227]	
Performance status	**[228]**/[229]	**[228]**/[229]		
Functioning tumor	[230]			
Non-functioning tumor				
No Diabetes	** [231] **			
No concomitant metformin use	** [231] **			
**Laboratory test/Biological marker**				
CgA, high	[228,232,233]	**[90,228]**/[232]		
No early CgA response	[228]	[228]		
NSE, high	** [228] **	**[90]**/[228]		
No early CgA or NSE response	** [228] **			
NLR, high	[234]			
LMR, low	[234]			
PlGF, high		** [90] **		
sVEGFR1, high		[90]		
sVEGFR2, low				[235]
VEGFR3 SNP				[236]
SDF-1α, high			[235]	[235]
IL-6, high				** [236] **
Osteopontin, high			** [236] **	
Triglyceride, high (During first 3 mo.)	[237]			
Hypercholesterolemia (Grade2), no	[232]	[232]		
**Imaging factor**				
SRS, negative				
SRS, asphericity	** [238] **			
**Treatment-related factor**				
Non-responder		[94]	**[239]**/[240,241]	[242]
Disease-progression (at 3/6/12 month)		[175]		
No primary tumor resection	** [231] **	[232]		
No resection, post-treatment				
Prior systemic therapy	[229]	**[229]**/[243]		
No prior PRRT	** [238] **			
Dose intensity, low		[243]		
Cumulative dose, low	** [243] **	[243]		
Stomatitis (within 8 week), no	[244]			
**Histological factor/classification/grading**				
Grade	**[95,231]**/[230,232]			
Ki-67 index			**[236]**/[227]	**[236]**/[227]
Mitosis			[227]	[227]
Differentiation (non-well/poorly)		** [228] **	** [227] **	** [227] **
Number of metastatic sites	[232,238]	[232]		
Lymph node metastasis				
Liver metastasis	** [231] **			
Lung metastasis				
Bone metastasis	[232,238]	[232]		
Hepatic tumor burden				
**Molecular factor**				
ACC1 high	[237]			
PHLDA-3, positive	[229]	[229]		
phospho-p70S6K, high	[232]	[232]		
Synaptophysin, negative			[227]	

Results of a multi-variate analysis are shown in red and bold square brackets to the left of /, while results of uni-variate analysis are shown to the right of. ACC1, acetyl-CoA carboxylase 1; CgA, chromogranin A; IL-6, interleukin-6; LMR, Lymphocyte-To-monocyte ratio [NSE, neuron-specific enolase; OS, overall survival; PFS, progression-free survival; PlGF, placental growth factor; SDF-1α, stromal cell-derived factor 1α; SNP, single nucleotide polymorphism; VEGF, vascular endothelial growth factor.

**Table 4 cancers-14-01250-t004:** Factors associated with worse PFS and OS with chemotherapy in advanced panNENs.

Factors	PFS	OS
**Clinical factor**		
Age	** [309] **	**[304,310,311]**/[305]
Male	[312]	
Performance status	** [302,303,307,308] **	**[298,304,307,308]**/[305,306]
Functioning tumor	**[302]**/[312]	
Non-functioning tumor		[311]
**Laboratory test/Biological marker**		
CgA response, no	[305,306,314]	[315]
5-HIAA, high	[305]	
NLR, high	[316]	
PLR, high	[316]	
LDH, high		**[298]**/[308]
ALP high	[317]	
CRP, high	[317]	
Platelet, high		** [298] **
Albumin, low	[317]	
**Imaging factor**		
SRS, negative		[304]
Tumor growth rate, high	** [319] **	** [319] **
**Treatment-related factor**		
Non-responder	[306,313,314]	[306]
No primary tumor resection	**[302]**/[313]	[306,312]
Prior Chemotherapy	**[229,302,310,318]**/[303,311,317]	**[229,302,310,311,318]**/[229]
Treatment cycles	** [317] **	
**Histological factor/classification/grading**		
Grade	**[309,312,319]**/[303,316,320]	**[312,319]**/[318]
Ki-67	**[302,303,305,306,318,321,322,323]**/ [310,324]	**[304,305,312,318,321]**/ [295,298,310,323,324]
Differentiation, poorly	**[317]**/[299,319,323,324]	**[307]**/[295,299,319,323,324,325,326]
Stage	**[312]**/[316]	
Size		[308]
Number of organs involve		** [321] **
Hepatic tumor burden		[321]
Extrahepatic metastasis	** [302,303] **	** [307] **
**Molecular factor**		
ALT negative		[327]
KRAS mutant		[295]
MGMT methylation, low	**[318,328]**/[229,329]	**[318,328]**/[229]
MGMT expression, high	**[328]**/[329]	**[328]**/[330]
Rb loss		** [295] **
Thymidylate synthase, deficient	[305]	

Results of multi-variate analysis are shown in red and bold square brackets to the left of /, while results of univariate analysis are shown to the right of /. ALT, alternative lengthening of telomeres; CgA, chromogranin A; HIAA, hydroxyindoleacetic acid; KRAS, Kirsten rat sarcoma viral oncogene homolog; LDH, lactate dehydrogenase; MGMT, O [6]-methylguanine-DNA methyltransferase; NLR, neutrophil-to-lymphocyte ratio; OS, overall survival; PFS, progression-free survival; PLR, platelet-to-lymphocyte ratio; Rb; retinoblastoma; SRS, somatostatin receptor scintigraphy; SUV, standardized uptake value.

**Table 5 cancers-14-01250-t005:** Factors associated with worse PFS and OS with liver-directed therapy in advanced panNENs.

Factors	PFS	OS
**Clinical factor**		
Age		**[347]**/[348]
Performance status		**[349,350]**/[351]
**Laboratory test/Biological marker**		
CgA, high		[347]
CgA increase after TACE	[356]	[356]
Pancreastatin, high		**[342]**/[352]
Bilirubin, high		[353]
ALP, high		**[354]**/[355]
Child-Pugh class, B or C		**[351]**/[355]
**Imaging factor**		
No ETB response		** [350] **
Lung shunt fraction >10%		[353]
Arteriovenous shunt		** [361] **
**Treatment-related factor**		
Non-responder		**[348,357]**/[347,351]
Number of TACE sessions		**[347]**/[352]
No primary tumor resection		[347,362,363]
Enterobiliary communication		** [358] **
Prior systemic therapy	**[348]**/[358,359]	[348,353,355,358]
No concomitant SSA use	** [348] **	**[348]**/[356]
No adjuvant therapy	[349]	
No surgery for metastasis		[347]
**Histological factor/classification/grading**		
Grade	**[348,349,359]**/[356]	**[342,347,348,349,357]**/[351,354,356]
Ki-67		[363,364]
Differentiation, poorly	** [361] **	[363,365]
Size		[365]
Hepatic tumor burden	** [348,349,361] **	**[342,347,348,349,357,358,361,365]**/[352,354]
Extrahepatic metastasis	** [358] **	**[355,357,358]**/[347,348,350,361,362,365]
Lymph node metastasis		** [351] **
Ascites		** [355] **
Portal vein thrombosis		** [350] **
**Molecular factor**		
DAXX, mutant	** [359] **	
MEN1, mutant	[359]	

Results of the multi-variate analysis are shown in red and bold square brackets to the left of /, while results of univariate analysis are shown to the right of.

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
