# Peer review of "Predictive Factors for Resistant Disease with Medical/Radiologic/Liver-Directed Anti-Tumor Treatments in Patients with Advanced Pancreatic Neuroendocrine Neoplasms: Recent Advances and Controversies"

_cancers, 2022, doi:10.3390/cancers14051250_

Round 1

Reviewer 1 Report

This review paper was very well written according to correct evaluation of previous papers, especially on critical review points by authors. Despite vast majority of data in the treatment for NEN, especially pancreatic NEN, authors reviewed these studies on the base of its role making evidence in many previously published literatures, Therefore it is very clearly understood to how each paper played a role for making evidence in this field.

1, In the sessions 9 and 10, there were some overlapping descriptions about the predictive factors of therapeutic response. Therefore, authors should revise the session 9 and 10 for deleting redundant descriptions for shortening these summarized sessions.

2, The title of session 9 and 10 might be not appropriate also with its content.

These titles should be changed to more easily understandable one.

Author Response

Point by point response to reviewer 1

Reviewer-1 #1. In the sessions 9 and 10, there were some overlapping descriptions about the predictive factors of therapeutic response. Therefore, authors should revise the session 9 and 10 for deleting redundant descriptions for shortening these summarized sessions.

Reply. We have edited sections 9 and 10 as recommended to reduce any overlap.  

Reviewer 1-#2. The title of session 9 and 10 might be not appropriate also with its content.

These titles should be changed to more easily understandable one.

Reply. We have revised the titles of  sections 9 and 10 as recommended to make it clearer.

Reviewer 2 Report

The submitted manuscript by Lee is a review paper on 'Predictive Factors for Resistant Disease with Medi-2 cal/Radiologic/Liver-Directed Anti-Tumor Treatments' in PanNEN patients. The article is written in great detail. Therefore, overlap within sections should be reduced and some passages shortened. It would be sufficient for the reader to focus on SSA, TKI, and PRRT and omit liver-specific therapy. 

Some Comments:
- The lead up to the question has been presented in great detail in the introduction. A somewhat more stringent presentation would be desirable.
- Please delete in line 53 etc or elaborate
- Table 1 contains the most relevant questions, from my point of view very good - please adjust indented sentences
- according to the statements in the method part, that one focuses on studies of the last 3-5 years, e.g. under 3.1 the side effects of everolimus can be omitted and generally check again critically less relevant data to be deleted
- The concept of table 2 is visually not ideal, reference 145 has also a blue background times? Similar applies to table 3/4.
- Figure 1: Abbreviations should be mentioned under the figure. In figure 1, the core statement is not quite clear to me, too many brackets are used. A clearer presentation would be beneficial for the reader.
- partly there are overlaps between the introduction and the main part, concerning the study presentation
- in principle, I would rather start chronologically with SSA, TKI and then PRRT
- I would leave out liver-directed therapy because of the longer review

Author Response

Point by point response to reviewer 2

Reviewer-2 ;  overall comment. The submitted manuscript by Lee is a review paper on 'Predictive Factors for Resistant Disease with Medi-2 cal/Radiologic/Liver-Directed Anti-Tumor Treatments' in PanNEN patients. The article is written in great detail. Therefore, overlap within sections should be reduced and some passages shortened. It would be sufficient for the reader to focus on SSA, TKI, and PRRT and omit liver-specific therapy. 

Reply.

  1. We have reread the paper and reduced overlap between sections, especially section 9 and 10. The other sections deal with separate subjects so there was not much overlap in them, but we have also removed any we found.
  2. We have completely revised the order of the sections as recommended, putting SSA first, followed by molecular therapies (TKI/mTor inhibitors, then PRRT, chemotherapy.
  3. For completeness we would like to retain the liver-directed therapy section, because this is increasingly used by a number of groups and the relationship of its sequence of use to the other treatment modalities is particularly unclear. A number of the recent predictive factors discussed here may help resolve some of this uncertainty in its place of use and therefore would be important to include.

Reviewer 2. Minor point #1. The lead up to the question has been presented in great detail in the introduction. A somewhat more stringent presentation would be desirable

Reply. We have edited the introduction to make it more focused.

Reviewer 2. Minor point #2. Please delete in line 53 etc or elaborate

Reply. We have revised this as recommended.

Reviewer 2. Minor point #3. Table 1 contains the most relevant questions, from my point of view very good - please adjust indented sentences

Reply. We have adjusted the indented sentences.

Reviewer 2. Minor point #4.According to the statements in the method part, that one focuses on studies of the last 3-5 years, e.g. under 3.1 the side effects of everolimus can be omitted and generally check again critically less relevant data to be deleted

Reply. We have removed the section on the side-effects of everolimus as recommended.

Reviewer 2. Minor point #5.The concept of table 2 is visually not ideal, reference 145 has also a blue background times? Similar applies to table 3/4.

Reply. Tables 2-5 are meant to clearly summarize the data in the literature for both multivariate and univariate analysis so the reader can go directly to the individual studies. We tried other formats, but all took much more space and multiple different Tables were required, so we thought this the best method of presenting the data directly succinctly.

            We have removed the blue color from ref 145.

Reviewer 2. Minor point #6. Figure 1: Abbreviations should be mentioned under the figure. In figure 1, the core statement is not quite clear to me, too many brackets are used. A clearer presentation would be beneficial for the reader.

.

Reply. 1. We have defined all abbreviations in the figure legend.

  1. We have removed all brackets
  2. We have rearranged the order of the treatments as requested
  3. We have revised the Figure to make it clear that this shows the percentage resistance to the indicated treatment in the indicated study.

Reviewer 2. Minor point #7.  Partly there are overlaps between the introduction and the main part, concerning the study presentation.

Reply. 1. We have edited the paper both in the introduction and sections 9 and 10 to reduce overlap.

Reviewer 2. Minor point #8.  In principle, I would rather start chronologically with SSA, TKI and then PRRT

Reply. 1. We have completely revised the paper including the Figure to the order the reviewer recommends which is: somatostatin analogue data, molecular therapy (everolimus, TKI), PRRT and chemotherapy

Reviewer 2. Minor point #9.  I  would leave out liver-directed therapy because of the longer review

Reply. For completeness we would like to retain the liver-directed therapy section, because this is increasingly used by a number of groups and the relationship of its sequence of  use to the other treatment modalities is particularly unclear. A number of the recent predictive factors discussed here may help resolve some of this uncertainty in its place of use and therefore would be important to include.

Round 2

Reviewer 2 Report

Some points have been revised adequately, some points discussed. Overall, the given review is too long with appr. 400 references and not easy to read.